# Horseshoe crab genomes reveal the evolution of genes and microRNAs after three rounds of whole genome duplication

Wenyan Nong [1,19], Zhe Qu [1,19], Yiqian Li [1,19], Tom Barton-Owen [1,19], Annette Y. P. Wong [1,19], Ho Yin Yip [1], Hoi Ting Lee [1], Satya Narayana [1], Tobias Baril [2], Thomas Swale [3], Jianquan Cao [1], Ting Fung Chan [4], Hoi Shan Kwan [5], Sai Ming Ngai [4], Gianni Panagiotou [6,7], Pei-Yuan Qian [8], Jian-Wen Qiu [9], Kevin Y. Yip [10], Noraznawati Ismail [11], Siddhartha Pati [12,13,14], Akbar John [15], Stephen S. Tobe [16], William G. Bendena [17], Siu Gin Cheung [18], Alexander Hayward [2] & Jerome H. L. Hui [1✉]

Whole genome duplication (WGD) has occurred in relatively few sexually reproducing invertebrates. Consequently, the WGD that occurred in the common ancestor of horseshoe crabs ~135 million years ago provides a rare opportunity to decipher the evolutionary consequences of a duplicated invertebrate genome. Here, we present a high-quality genome assembly for the mangrove horseshoe crab *Carcinoscorpius rotundicauda* (1.7 Gb, N50 = 90.2 Mb, with 89.8% sequences anchored to 16 pseudomolecules, $2n = 32$), and a resequenced genome of the tri-spine horseshoe crab *Tachypleus tridentatus* (1.7 Gb, N50 = 109.7 Mb). Analyses of gene families, microRNAs, and synteny show that horseshoe crabs have undergone three rounds (3R) of WGD. Comparison of *C. rotundicauda* and *T. tridentatus* genomes from populations from several geographic locations further elucidates the diverse fates of both coding and noncoding genes. Together, the present study represents a cornerstone for improving our understanding of invertebrate WGD events on the evolutionary fates of genes and microRNAs, at both the individual and population level. We also provide improved genomic resources for horseshoe crabs, of applied value for breeding programs and conservation of this fascinating and unusual invertebrate lineage.

A list of author affiliations appears at the end of the paper.

Polyploidy provides new genetic raw material for evolutionary diversification, as gene duplication can lead to the evolution of new gene functions and regulatory networks[1]. Nevertheless, whole-genome duplication (WGD) is a relatively rare occurrence in animals when compared to fungi and plants[2]. Two rounds of ancient WGD occurred in the last common ancestor of the vertebrates, with additional rounds in some teleost fish lineages[2–4]. Fixation of these WGD events (i.e., 'polyploidization') is considered a major force in shaping the evolutionarily success of vertebrate lineages, by facilitating fundamental changes in physiology and morphology, leading to the origin of new adaptations[5,6]. Among the invertebrates, horseshoe crabs[7–9], spiders, and scorpions[10] represent the only sexually reproducing lineages that are known to have undergone WGD (Fig. 1a).

Horseshoe crabs are considered to be 'living fossils'. The oldest actual fossils of horseshoe crabs date to the Ordovician period ~450 million years ago (Mya)[11], and remarkably, extant species remain relatively unchanged morpologically since this extremely ancient date. However, despite their long history, there are only four extant species of horseshoe crabs worldwide: the Atlantic horseshoe crab (*Limulus polyphemus*) from the Atlantic East Coast of North America, and the mangrove horseshoe crab (*Carcinoscorpius rotundicauda*), the Indo-Pacific horseshoe crab (*Tachypleus gigas*), and the tri-spine horseshoe crab (*Tachypleus tridentatus*), from South and East Asia[12]. All extant horseshoe crabs are estimated to have diverged from a common ancestor that existed ~135 Mya[13], and they share an ancestral WGD[9]. A high-quality genome assembly was recently announced as a genomic resource for *T. tridentatus*[14,15], leaving an exciting research opportunity to analyse the genomes of other horseshoe crab species to understand how WGD events reshape the genome and rewire genetic regulatory networks in invertebrates.

In the present study, we provide the first high quality genome of the mangrove horseshoe crab (*C. rotundicauda*), and a resequenced genome of the tri-spine horseshoe crab (*T. tridentatus*). Importantly, we present evidence for the number of rounds of WGD that have occurred in these genomes, and investigate if these represent a shared event with spiders. We also examine the evolutionary fate of genes and microRNAs at both the individual and population level in these genomes. Collectively, this study highlights the evolutionary consequences of a unique invertebrate WGD, while at the same time providing detailed genetic insights of utility for diverse genomic, biomedical, and conservation applications.

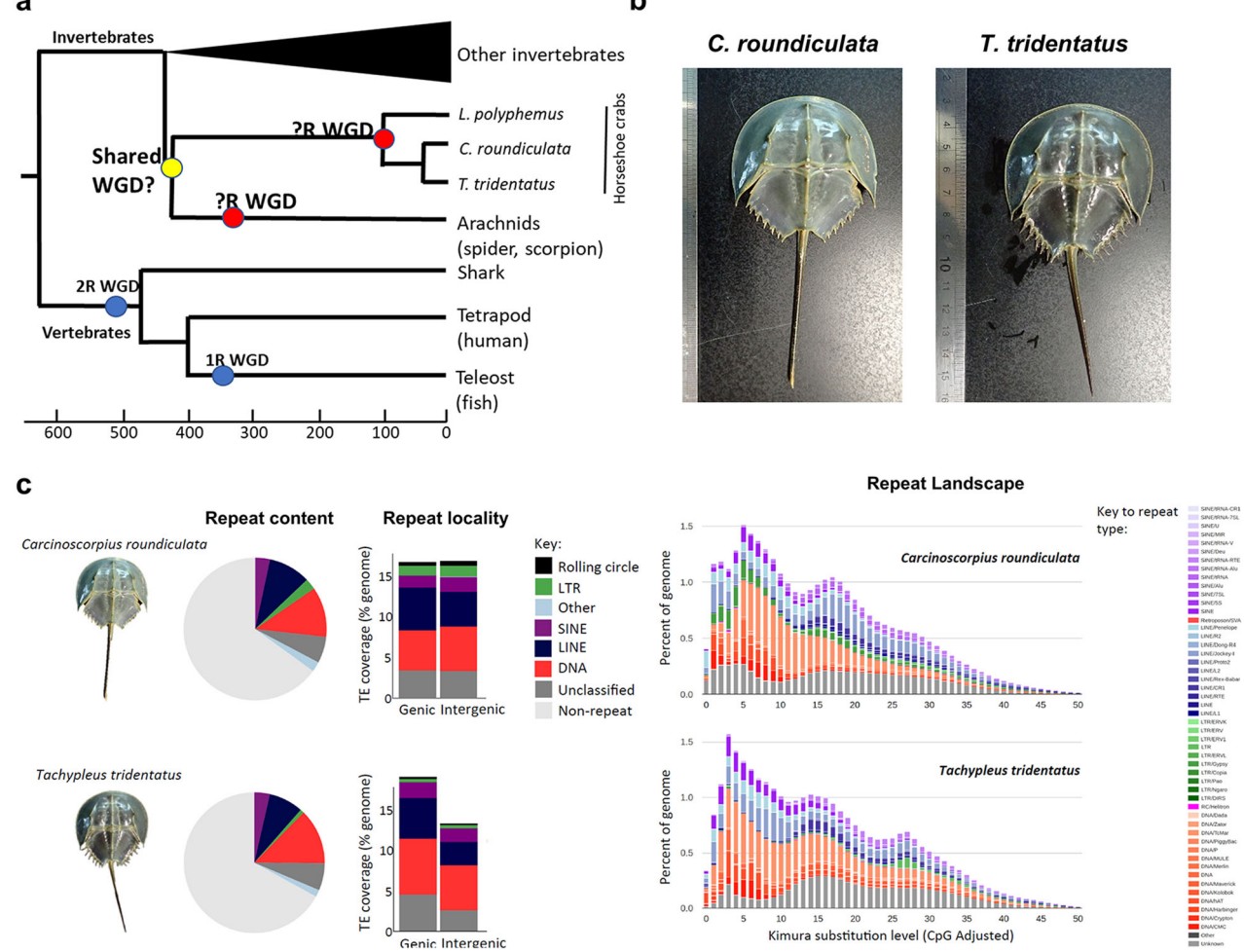

**Fig. 1 Horseshoe crabs C. roundicultata and T. tridentatus. a** Schematic diagram illustrating the current knowledge of whole-genome duplication (WGD) in animals. '?R' denotes unknown rounds of whole-genome duplication; **b** pictures of horseshoe crabs *C. roundicultata* and *T. tridentatus*; **c** Repeat content for the two horseshoe crab genomes, *C. rotundicauda* and *T. tridentatus*: Pie charts illustrating repeat content as a proportion of total genomic content; Repeat content present in genic verses intergenic regions; and Repeat landscape plots illustrating transposable element activity in each horseshoe crab genome. Source data reveals these figures can be found in Supplementary Data 8.

## Results and discussion

**High-quality genomes of two horseshoe crabs**. Genomic DNA was extracted from single individuals of two species of horseshoe crab, C. rotundicauda and T. tridentatus (Fig. 1b), and sequenced using Illumina short-read, 10X Genomics linked-read, and PacBio long-read sequencing platforms (Supplementary Tables 1–4). Hi–C libraries were also constructed for both species sequenced using the Illumina platform (Supplementary Figs. 1 and 2). For the final genome assemblies, both genomes were first assembled using short-reads, followed by scaffolding with Hi-C data. The C. rotundicauda genome assembly is 1.72 Gb with a scaffold N50 of 90.2 Mb (Table 1). The high physical contiguity of the genome is matched by high completeness, with 93.8% complete BUSCO core eukaryotic genes (Table 1). The T. tridentatus genome is 1.72 Gb with a scaffold N50 of 109.7 Mb and 93.7% BUSCO completeness (Table 1). In total, the C. rotundicauda and T. tridentatus genome assemblies include 34,354 and 42,906 gene models, respectively. Furthermore, 89.8% of the sequences assembled for C. rotundicauda genome are contained on just 16 pseudomolecules, consistent with a near chromosome-level assembly (chromosome $2n = 32$,[16], Supplementary Table 3).

To date, the only repeat data available for horseshoe crabs are two independent analyses of the tri-spine horseshoe crab T. tridentatus, which identified repeat contents of 34.61%[14], and 39.96%[15]. In the present study, we provide the first comparative analysis of repeat content in horseshoe crab genomes, by analysing repeats in our genome assembly for T. tridentatus, as well as our assembly for the mangrove horseshoe crab, C. rotunicauda. We find that repeat content is similar in both genomes, occupying approximately one third of total genomic content. Specifically, we identify a total repeat content of 32.99% for T. tridentatus and 35.01% for C. rotunicauda, of which the dominant repeats are DNA elements, followed by LINEs, with SINEs and LTR elements contributing just a small proportion of total repeat content (Fig. 1c, Supplementary Table 5).

A large proportion of eukaryotic genomes is typically composed of repetitive DNA, and repeats are widely cited as being one of the key determinants of genome size[17]. However, while the genome size for both species of horseshoe crab sequenced here is comparatively large for invertebrate genomes, their repeat content is not unusually high (i.e., C. rotundicauda: 35.01%, T. tridentatus: 32.99%, Fig. 1c, Supplementary Table 5, Supplementary Note). Instead, the comparatively large size of horseshoe crab genomes appears to be a consequence of multiple rounds of WGD, as discussed in greater detail below.

In the C. rotundicauda genome, repeats are evenly distributed across genic and intergenic regions (Fig. 1c). However, in the T. tridentatus genome, a greater proportion of repeats are found in genic regions, due primarily to a higher density of DNA elements and LINEs, as well as unclassified elements in these regions (Fig. 1c). Repeat landscape plots (Fig. 1c) suggest a relatively similar pattern of historical transposable element activity for both horseshoe crab species. However, recent transposable element activity appears to have tapered off more quickly in the T. tridentatus genome, particularly with respect to LTR elements and certain DNA elements (Fig. 1c).

**Three rounds (3R) of whole-genome duplications in horseshoe crabs**. Initial efforts to analyse WGD in extant horseshoe crabs used low-depth and genotyping-by-sequencing which did not provide sufficient resolution to understand the consequences of WGD in these taxa[7–9]. Recently, there were two resequencing efforts for the horseshoe crab T. tridentatus[14,15], but our T. tridentatus genome assembly has the largest contig N50 (Table 1). Furthermore, our assembly for C. rotundicauda represents the

**Table 1 Summary of genome assembly statistics of horseshoe crabs.**

| Common name | Mangrove horseshoe crab | | Tri-spine horseshoe crab | | | Atlantic horseshoe crab | |
|---|---|---|---|---|---|---|---|
| Species name | Carcinoscorpius rotundicauda | | Tachypleus tridentatus | | | Limulus polyphemus | |
| Accession number | WCHO00000000 | GCA_011833715.1 | WCHN00000000 | CNA0000821 | QXHF01000000 | GCF_000517525 | PRJNA187356 |
| Number of scaffolds | 30,138 | 728 | 39,367 | 204 | 671,877 | 286,793 | 896,522 |
| Assembly size | 1,725,596,044 | 1,668,682,064 | 1,718,441,268 | 2,167,470,406 | 1,942,936,674 | 1,828,271,751 | 1,229,280,963 |
| Gap content | 0.85% | 0.05% | 3.10% | 0.17% | 1.55% | 6.70% | 12.65% |
| Scaffold N50 | 90,264,435 | 102,344,000 | 109,788,719 | 169,002,194 | 2,761,313 | 254,089 | |
| Contig N50 | 437,918 | 7,608,042 | 2,961,265 | 1,689,442 | 52,179 | 11,441 | 2,929 |
| Number of genes | 34,354 | 25,985 | 42,906 | 34,966 | 29,134 | 26,905 | 460 |
| Complete BUSCOs | 93.80% | 94.80% | 93.70% | 95.00% | 96.20% | / | / |
| Reference | This study | [63] | This study | [14] | [15] | [8] | [7] |

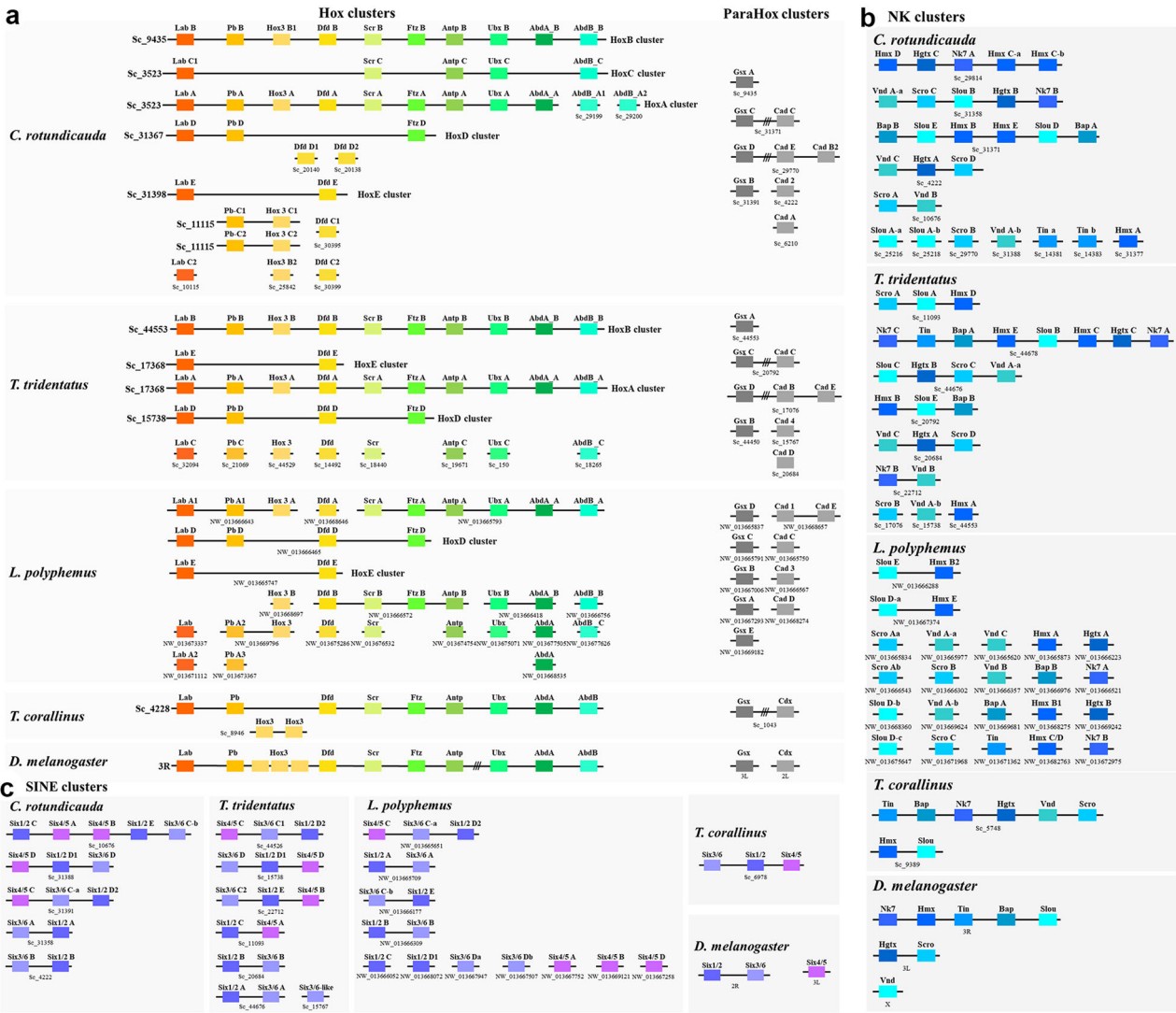

**Fig. 2 Homeobox gene organisation. a** Genomic organisation of the Hox (left) and ParaHox (right) cluster genes in the horseshoe crab genomes. **b** Genomic organisation of the NK and **c** SINE cluster genes in the horseshoe crab genomes. Note that the *L. polyphemus* genome assembly (GCF_000517525) was obtained from Battelle et al. [8].

first close to chromosomal-level genome assembly for this species. Consequently, the two high-quality horseshoe crab genomes presented in this study provide an unprecedented opportunity to address the issue of invertebrate WGD and its evolutionary consequences.

An important outstanding question is how many rounds of WGD occurred in the last common ancestor of horseshoe crabs, or alternatively, if all rounds of WGD occurred already in the ancestor of arachnids and horseshoe crabs (Fig. 1a)? To address this question, we first investigated the number and genomic location of *Hox* cluster genes, which have played the role of a 'Rosetta stone' for understanding animal evolution[18]. For example, the genome of the cephalochordate amphioxus contains only a single *Hox* gene cluster with 15 *Hox* genes, while the mouse genome contains four *Hox* gene clusters with 39 *Hox* genes, providing evidence that two rounds of WGD occurred between the most recent common ancestor of amphioxus and human[19,20]. In our horseshoe crab genomes for *C. rotundicauda* and *T. tridentatus*, the number of *Hox* genes was found to be 43 and 36, respectively (Fig. 2a, Supplementary Data 1). In *C. rotundicauda*, we found that there are five *Hox* clusters, with other *Hox* genes located on additional small scaffolds; while in *T. tridentatus*, there

are three *Hox* clusters, again with other *Hox* genes scattered across different scaffolds (Fig. 2a). This situation is similar to that for the genome assembly of the Atlantic horseshoe crab *L. polyphemus*[8], where our analyses demonstrated that there are four *Hox* clusters with additional *Hox* genes located on different scaffolds. In a recent study of the *T. tridentatus* re-sequenced genome, the authors could only detect two *Hox* clusters, and could not identify the *Ftz* gene inside these clusters[14]. In contrast, our results suggest that there are in fact three *Hox* clusters (including *Ftz*), and thus more than one round of WGD occurred in the lineage leading to extant horseshoe crabs.

We next investigated the sister cluster of the *Hox* genes—the *ParaHox* cluster genes, which are also highly clustered in bilaterians[21–23]. Similar to the *Hox* cluster genes, the cephalochordate amphioxus contains only a single *ParaHox* gene cluster in its genome, while the *ParaHox* cluster genes are located on four chromosomes in human[19]. In comparison, both the horseshoe crab genomes for *C. rotundicauda* and *T. tridentatus* contain two *ParaHox* clusters, composed of *Gsx* and *Cdx*, with other *ParaHox* genes located on three additional scaffolds. Meanwhile, in the genome assembly of *L. polyphemus*[8], perhaps due to the lower sequence continuity of the genome (i.e. low

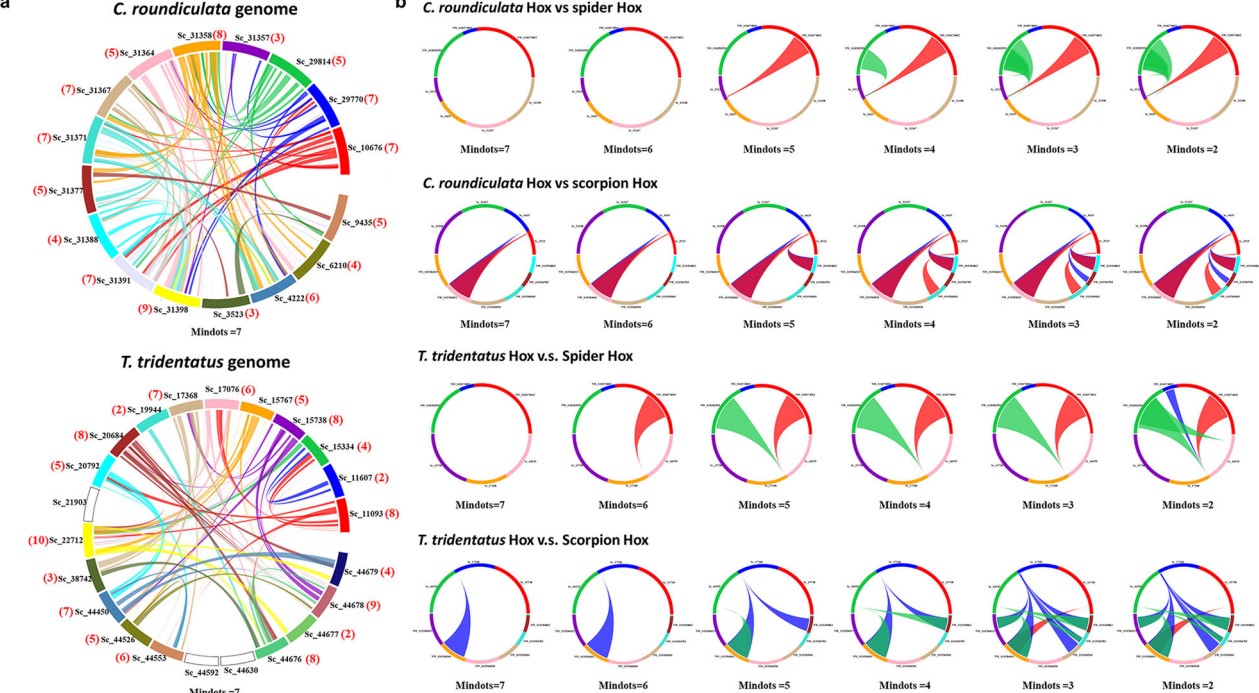

**Fig. 3 Syntenic regions between chelicerate genomes. a** Synteny between different chromosomes of *C. roundiculata* and *T. tridentatus*. Note that the bracketed numbers highlighted in red refer to the numbers of chromosomes that syntenic blocks with that chromosome (counting include its own copy). **b** Synteny relationships of Hox scaffolds of (Upper panel): *C. roundiculata*, spider and scorpion; (Lower panel) and *T. tridentatus*, spider, and scorpion.

scaffold N50), only a single *ParaHox* cluster for *Cdx* was identified, with the other *ParaHox* genes located on eight additional scaffolds (Fig. 2a). For other well-known homeobox gene clusters, including the *NK* cluster and *SINE* clusters, multiple clusters were revealed, as above (Fig. 2b, c). In *C. rotundicauda* and *T. tridentatus*, five and seven *SINE* clusters were found respectively, while in the genome assembly of *L. polyphemus*[8], four *SINE* clusters were revealed, with the other six genes located elsewhere in the genome.

Using genome-wide analyses of homeobox gene content in three horseshoe crab genomes, we find that many homeobox genes are present in more than four copies (details are provided in Supplementary Table 6, Supplementary Figs. 3–7). These results suggest that at least two rounds (2R), and likely three rounds (3R) of WGD have occurred. So the question then becomes, exactly how many rounds of WGD did occur in horseshoe crab genomes? To address this question, we carried out additional genome-wide synteny analyses. We found that, using a default set of a minimum of seven genes to define a syntenic block, most of the chromosomes of *C. rotundicauda* exhibit synteny with on other chromosomes (between 4–8 including its own copy) (Fig. 3a). Thus, we propose that three rounds of whole-genome duplication (3R WGD) occurred in horseshoe crabs.

**Shared or independent duplications with spiders?** Another major unresolved question relating to horseshoe crab genomes is whether the reported cases of WGD in chelicerates constitute shared or independent events. Gene family analyses of spider and scorpion genomes have suggested that an ancient WGD is shared between them, independent of the further WGDs that occurred in the horseshoe crab lineage[10]. Using the two horseshoe crab genome assemblies generated here, this study addressed this important question from two different perspectives by: (1) performing analyses of synteny, and (2) considering recent new evidence on phylogenetic relationships within the Chelicerata.

We first carried out analyses of synteny between the *Hox* scaffolds of *C. rotundicauda* and published spider and scorpion genomes[10] (Fig. 3b). Despite no clear shared duplication event between *C. rotundicauda* and spider *Hox* clusters, surprisingly, we observed syntenic relationships between two Hox scaffolds when using a minimum of five genes to define a syntenic block (Fig. 3b). Similarly, during synteny comparisons of *Hox* scaffolds of *T. tridentatus* and published spider and scorpion genomes, we observed syntenic relationships between two different *Hox* scaffolds when using a minimum of five genes to define a syntenic block (Fig. 3b). Under less stringent criteria using a minimum of two genes to define a syntenic block, we additionally observed syntenic relationships for two other *Hox* scaffolds between *T. tridentatus* and spider genomes (Fig. 3b).

An important consideration necessary to fully understand WGD events identified for horseshoe crab and chelicerate genomes is the phylogenetic relationships between these lineages. Horseshoe crabs have long been regarded as a monophyletic group (Xiphosura) and the sister group to the terrestrial chelicerate clade that includes spiders, scorpions, ticks, mites, harvestmen, and solifuges (Arachnida). However, in a recent phylogenetic analysis using publicly available data, including three xiphosurans, two pycnogonids, and 34 arachnids, it was suggested that the horseshoe crabs represent a group of marine arachnids[24]. Conversely, another group of researchers recovered the Xiphosura as the sister group to the Arachnida[25], suggesting a single terrestrialisation event occurred after the last common ancestor of arachnids and horseshoe crabs diverged. Consequently, additional analyses and data are needed to differentiate between these scenarios, and fully demonstrate whether a shared WGD event occurred at the common ancestor of horseshoe crabs, spiders, and scorpions; or if an ancestral WGD occurred in the ancestral lineage to chelicerates and xiphosurans, followed by massive gene losses in some lineages, such as ticks and mites; or multiple WGDs originated independently in and arachnopulmonates and xiphosurans.

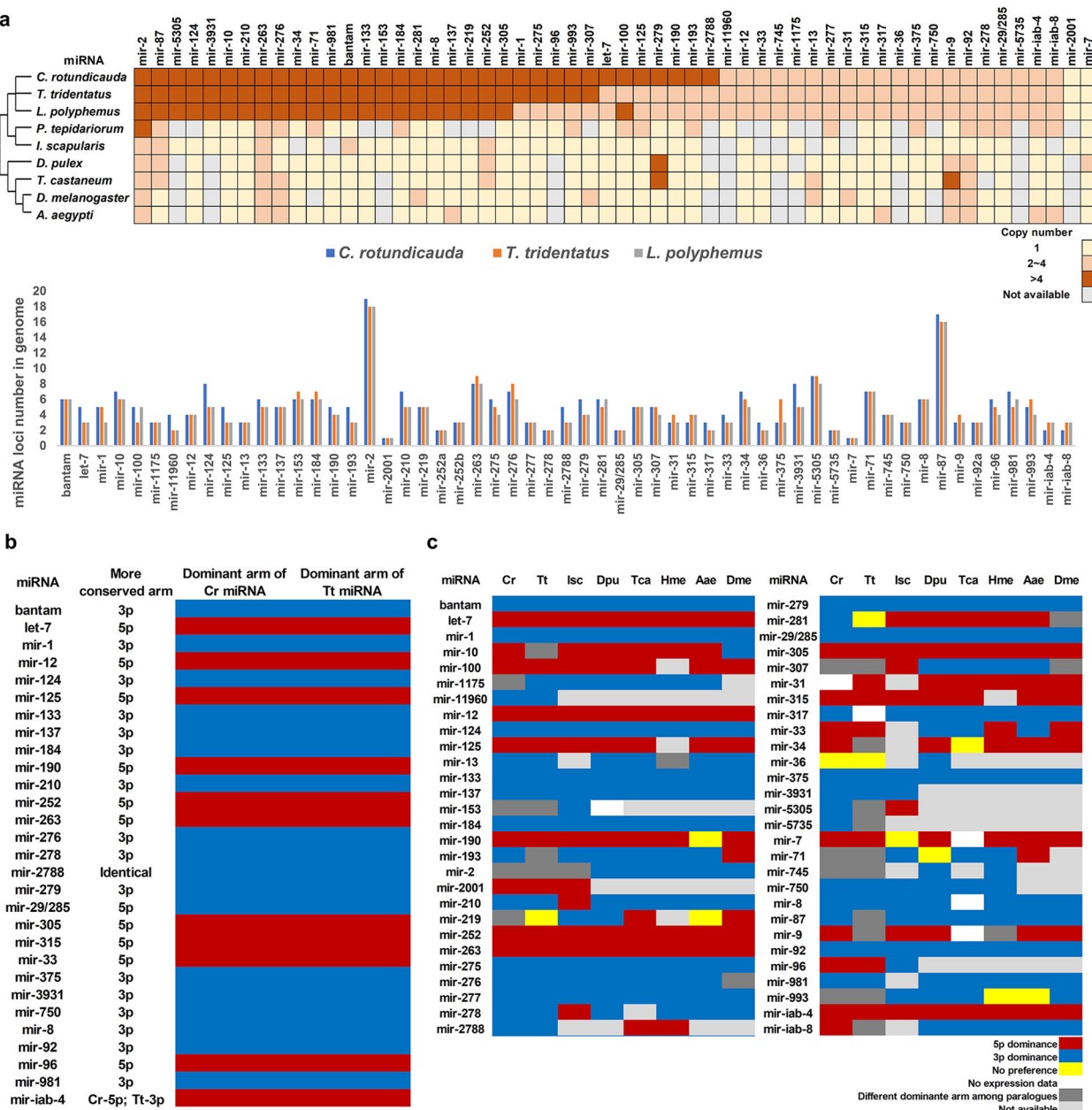

**Fig. 4 MicroRNA copies and dominant arm usage. a** Number of gene copies of conserved microRNAs in the arthropod genomes; **b** Sequence conservation and arm switching of horseshoe crab microRNAs; **c** Comparison of microRNA arm preference among different arthropod species. Isc *Ixodes scapularis*, Dpu *Daphnia pulex*, Tca *Tribolium castaneum*, Hme *Heliconius melpomene*, Aae *Aedes aegypti*, Dme *Drosophila melanogaster*. Arm preference: blue—3p dominance, red—5p dominance, yellow—no preference, white—no expression. Source data reveals Fig. 4a can be found in Supplementary Data 8.

**Duplicated fates of noncoding microRNAs.** The availability of new transcriptomic data, especially the first small RNA transcriptomic data for both species of horseshoe crabs (Supplementary Tables 7 and 8), enabled us to analyse the evolutionary consequences of small noncoding RNAs after the WGD events in both *C. rotundicauda* and *T. tridentatus*. To reveal if duplicated microRNAs can also provide insights into the number of rounds of WGD that occurred, we first examined the number of paralogues for the bilaterian conserved set of 57 microRNAs, across three horseshoe crab genomes (Fig. 4a, Supplementary Data 2–5). Of these microRNAs, 27 and 34 have more than 4 copies in *T. tridentatus* and *C. rotundicauda,* respectively (Fig. 4a, Supplementary Fig. 8). These data further support the hypothesis that 3R WGD occurred in horseshoe crabs.

To understand the fates of microRNA paralogues, we first analysed sequence divergence in 41 conserved microRNA families and 4 chelicerate-specific microRNAs by aligning their sequences (Supplementary Figs. 9–21, Supplementary Data 6). We found that the paralogues always have greater sequence conservation in one arm (rather than showing similar conservation for both arms across paralogues) after WGD (Supplementary Figs. 9–21). An example is illustrated for the microRNA *bantam*, where the sequence of the 5p arm is less conserved than the 3p arm between paralogues (Supplementary Fig. 22).

To explore whether greater conservation in the 3p microRNA arm correlates with a greater expression level, we mapped small RNA reads to different paralogues. By eliminating microRNA species which have different arm usage between their paralogues or between horseshoe crab species, we found that out of the 29

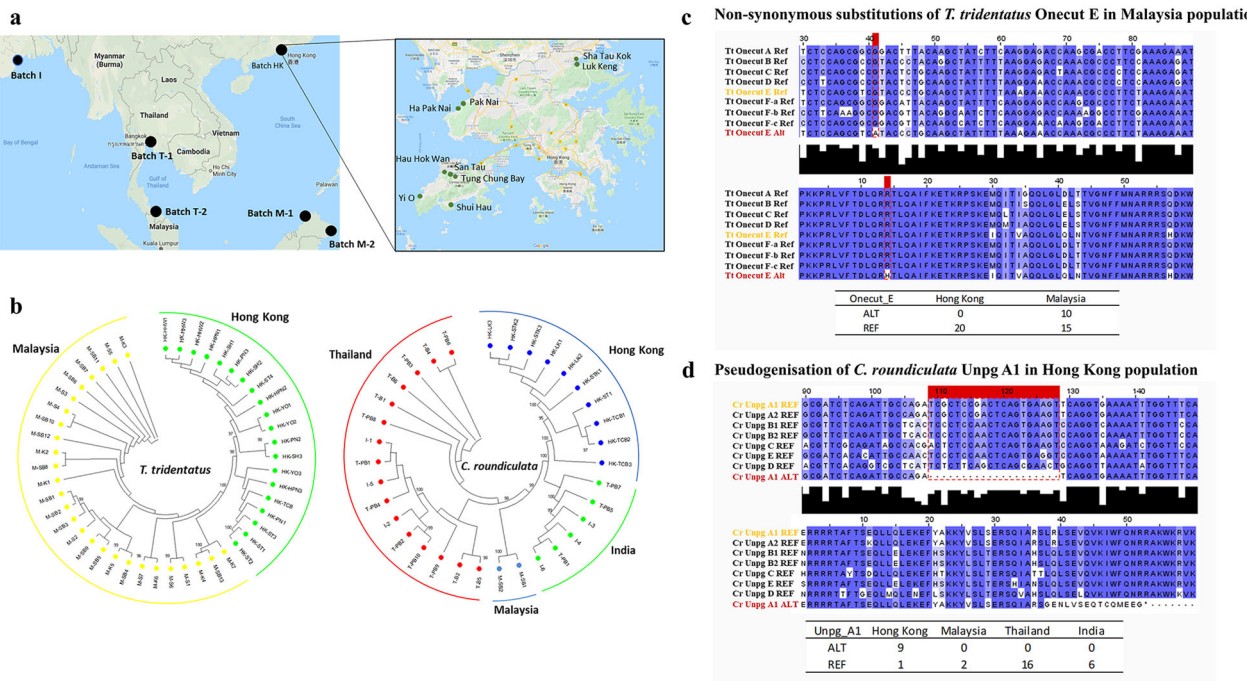

**Fig. 5 Population genomics and evolutionary fates of paralogues. a** Geographical distribution of *C. roundicultata* and *T. tridentatus* collected samples; **b** Phylogenetic trees of the collected samples. **c** Non-synonymous substitutions of *T. tridentatus* Onecut-E genes in individuals collected in Malaysia; **d** Pseudogenisation of *C. roundiculata* Unpg-A1 gene in individuals collected in Hong Kong population.

assessed microRNAs, 26 show a higher expression level/dominant arm usage for the conserved arm (Fig. 4b, Supplementary Data 2, 6). For example, the 3p arm shows more sequence conservation between *bantam* paralogues in horseshoe crabs, and they also show higher expression levels than corresponding 5p arms (Fig. 4b). The 26 conserved microRNAs identified as showing higher expression levels for the conserved arm serve as the first example correlating expression level and conservation of mature microRNA sequences in paralogues following WGD.

In addition to relatively ancient conserved microRNAs, we also investigated novel microRNAs specific to a certain horseshoe crab species, to understand the impact of WGD on these. We found that 7 xiphosuran novel miRNAs are conserved in all three horseshoe crab genomes, while another 13 miRNAs are conserved in both *C. rotundicauda* and *T. tridentatus* (Supplementary Figs. 23–25). Considering that these xiphosuran miRNAs paralogues are located on different scaffolds in their respective genomes, they are unlikely to be generated via tandem duplications. The identified novel microRNAs show higher sequence conservation between orthologues than paralogues (e.g., Supplementary Figs. 23–25), suggesting these horseshoe crab-specific novel microRNAs originate in the horseshoe crab ancestor following WGD.

In the common house spider *Parasteatoda tepidariorum*, which is believed to have undergone a single round of WGD[10], paralogues of microRNAs were found to exhibit arm switching, a phenomenon whereby dominant microRNA arm usage is swapped among different tissues, developmental stages or species[26,27]. We investigated microRNA arm switching in the sRNA transcriptomes generated here and compared this to their orthologues in various arthropods including fruitfly (*Drosophila melanogaster*), mosquito (*Aedes aegypti*), butterfly (*Heliconius melpomene*), beetle (*Tribolium castaneum*), water flea (*Daphinia pulex*), and tick (*Ixodes scapulari*)[28,29]. By comparing dominant arm usage across different species, we found that many microRNAs, such as *miR-2788*, *miR-281* and *miR-iab-8* have

undergone microRNA arm switching (Fig. 4c, Supplementary Data 2). Moreover, we also observed microRNA arm switching in cases of microRNAs throughout different developmental time or tissues (Supplementary Data 2). These findings are congruent with the spider microRNA study[10,27].

**WGD at the population level**. An additional question that remains poorly explored is the evolutionary consequences of WGDs on gene duplicates at the population level. To address this question, individuals of both *C. rotundicauda* and *T. tridentatus* were collected from different locations across Asia and subjected to genome sequencing (Fig. 5a, Supplementary Tables 9 and 10). We first mapped sequencing reads to the mitochondrial genome and constructed evolutionary trees from mitochondrial data to identify population structure. Distinct subpopulations were identified within different regions in Asia, for example, populations from Hong Kong formed a distinct group from other locations in Asia, which may be due to the strong ocean currents surrounding Hong Kong preventing gene flow between these locations (Fig. 5b, Supplementary Fig. 26).

Next, we sought to identify how dynamic mutations observed at paralogous genes are in different individuals. We focussed on homeobox genes, calling single-nucleotide polymorphisms (SNPs) at the homeodomains of all annotated homeobox genes and found confident cases of both non-synonymous substitutions as well as pseudogenisation in the homeodomain of certain populations (Fig. 5c; Supplementary Data 7). In *T. tridentatus*, non-synonymous substitutions at the homeodomain of *Six3/6-like* and *Onecut-E* genes were revealed in individuals from Malaysian populations (Fig. 5c, Supplementary Fig. 27a). Similarly, for *C. roundiculata*, non-synonymous substitutions at the homeodomain of the *En-D* gene were also revealed in individuals from populations in Thailand (Supplementary Fig. 27b). This is the first evidence demonstrating that different gene duplicates are under different rates of mutation and selection at the individual level after WGD in invertebrates.

Importantly, unique pseudogenisation was discovered in the paralogue of *Unpg* in many individuals in the *C. rotundicauda* population located in Hong Kong (Fig. 5d). In 9 out of the 10 individuals sequenced from Hong Kong, an alternative form with a deletion in *Unpg-A1* was identified (Fig. 5d). Given that homeodomains are standardised as transcription factors with a sequence length of ~60–63 amino acids[20], the observed deletion suggests that in certain individuals, these genes are in the process of becoming pseudogenes. This is the first evidence demonstrating the ongoing and dynamic mutation rate of paralogues at the population level after WGD in invertebrates.

**Conclusion**. Despite its importance in evolution, the impacts of WGD remain understudied, particularly in invertebrates such as horseshoe crabs. This study provides evidence of the 3R WGD events in horseshoe crabs, shedding light on the evolutionary fates of genes and microRNAs at both the individual and population levels, as well as highlighting the genetic diversity of these amazing animals, with importance for understanding their evolution, genomics, and practical value for breeding programmes and conservation.

## Methods

**DNA, mRNA, and sRNA extraction and sequencing**. Genomic DNA of the horseshoe crabs *C. rotundicauda* and *T. tridentatus* was isolated from the leg muscle of a single individual in each case, using the PureLink Genomic DNA Kit (Invitrogen). In addition, different tissues were dissected and homogenised in Trizol reagent (Invitrogen), and total RNA was isolated following the manufacturers' instructions. Blood samples of both species of horseshoe crab were drawn by syringe and directly transferred into Trizol reagent for RNA extraction. For egg, 1st, 2nd and 3rd instars of *T. tridentatus*, whole individuals were used for RNA extraction. Extracted gDNA was subject to quality control using gel electrophoresis. Qualified samples were sent to Novogene and Dovetail Genomics for library preparation and sequencing. In addition, a Chicago library was prepared by Dovetail Genomics using the method described by Putnam et al[30]. Briefly, ~500 ng of high molecular weight gDNA (mean fragment length = 55 kb) was reconstituted into artificial chromatin in vitro and fixed with formaldehyde. Fixed chromatin was digested with DpnII, the 5′ overhangs filled in with biotinylated nucleotides, and free blunt ends were ligated. After ligation, crosslinks were reversed, and the DNA purified. Purified DNA was treated to remove biotin that was not internal to ligated fragments. The DNA was then sheared to ~350 bp mean fragment size and sequencing libraries were generated using NEBNext Ultra enzymes and Illumina-compatible adaptors. Biotin-containing fragments were isolated using streptavidin beads before PCR enrichment of each library. The libraries were sequenced on the Illumina HiSeq X platform. Dovetail HiC libraries were prepared as described previously[31]. Briefly, for each library, chromatin was fixed with formaldehyde in the nucleus and then extracted Fixed chromatin was digested with DpnII, the 5′ overhangs filled in with biotinylated nucleotides, and free blunt ends were ligated. After ligation, crosslinks were reversed and the DNA purified. Purified DNA was treated to remove biotin that was not internal to ligated fragments. The DNA was then sheared to ~350 bp mean fragment size and sequencing libraries were generated using NEBNext Ultra enzymes and Illumina-compatible adaptors. Biotin-containing fragments were isolated using streptavidin beads before PCR enrichment of each library. Details of the sequencing data can be found in Supplementary Tables 1 and 2.

Total RNA was subject to quality control using a Nanodrop spectrophotometer (Thermo Scientific), gel electrophoresis and analysis using the Agilent 2100 Bioanalyzer (Agilent RNA 6000 Nano Kit). High quality samples underwent library construction and sequencing at Novogene; polyA-selected RNA-Sequencing libraries were prepared using TruSeq RNA Sample Prep Kit v2. Insert sizes and the concentration of final libraries were determined using an Agilent 2100 bioanalyzer instrument (Agilent DNA 1000 Reagents) and real-time quantitative PCR (TaqMan Probe), respectively. Small RNA (<200 nt) was isolated using the mirVana miRNA isolation kit (Ambion) according to the manufacturer's instructions. Small RNA was dissolved in the elution buffer provided in the mirVana miRNA isolation kit (Thermo Fisher Scientific) and submitted to Novogene for HiSeq Small RNA library construction and 50 bp single-end sequencing. Detailed information for the sequencing data can be found in Supplementary Tables 7 and 8.

**Genome, mRNA transcriptome, and sRNA assembly and annotation**. To process the Illumina sequencing data, adaptors were trimmed and reads were filtered using the following parameters '-n 0.1 (i.e. removal if N accounted for 10% or more of reads) -l 4 -q 0.5 (i.e., removal if the quality value is lower than 4 and accounts for 50% or more of reads)'. FastQC was run for quality control[32]. If adaptor

contamination was identified, adapter sequences were removed using minion[33]. Adapter trimming and quality trimming was then performed with cutadapt v1.10[34]. For each species, k-mers of the Illumina PE library of 500 bp insert size were counted using DSK version 2.1.0 with $k = 25$[35], and estimation of genome size, repeat content, and heterozygosity were analysed based on a k-mer-based statistical approach using the GenomeScope webtool[36]. Kraken was used to estimate the percentage of reads that may results from contamination from bacteria[37]. Chromium WGS reads were separately used to make a de novo assembly using Supernova (v 2.1.1), with the parameter '–maxreads = 23,1545,066' for *C. rotundicauda*, and '–maxreads = 100,000,000' for *T. tridentatus*, respectively. The de novo assembly, shotgun reads, Chicago library reads, and Dovetail HiC library reads were used as input data for HiRise, a software pipeline designed for using proximity ligation data to scaffold genome assemblies[30]. An iterative analysis was conducted. First, Shotgun and Chicago library sequences were aligned to the draft input assembly using a modified SNAP read mapper (http://snap.cs.berkeley.edu). The separation of Chicago read pairs mapped within draft scaffolds was analysed by HiRise to produce a likelihood model for genomic distance between read pairs, and the model was used to identify and break putative misjoins, to score prospective joins, and to make joins above a threshold. After aligning and scaffolding Chicago data, Dovetail HiC library sequences were aligned and scaffolded following the same method. After scaffolding, shotgun sequences were used to close gaps between contigs.

Raw sequencing reads of the transcriptomes were pre-processed with quality trimmed by trimmomatic (version 0.33, with parameters 'ILLUMINACLIP: TruSeq3-PE.fa:2:30:10 SLIDINGWINDOW:4:5 LEADING:5 TRAILING:5 MINLEN:25')[38]. For the nuclear genomes, the genome sequences were cleaned and masked by Funannotate (v1.6.0, https://github.com/nextgenusfs/funannotate)[39], the softmasked assembly were used to run 'funannotate train' with parameters '–stranded RF–max_intronlen 350,000' to align RNA-seq data, ran Trinity, and then ran PASA[40]. The PASA gene models were used to train Augustus in 'funannotate predict' step following manufacturers recommended options for eukaryotic genomes (https://funannotate.readthedocs.io/en/latest/tutorials.html#non-fungal-genomes-higher-eukaryotes). Briefly, the gene models were predicted by funannotate predict with parameters '–repeats2evm–protein_evidence uniprot_sprot.fasta–genemark_mode ET–busco_seed_species arthropoda–optimize_augustus–busco_db arthropoda–organism other–max_intronlen 350000', the gene models predicted by several prediction sources including GeneMark[41], high-quality Augustus predictions (HiQ), PASA[40], Augustus[42], GlimmerHMM[43] and snap[44] were passed to Evidence Modeler[40] (EVM Weights: {'GeneMark': 1, 'HiQ': 2, 'pasa': 6, 'proteins': 1, 'Augustus': 1, 'GlimmerHMM': 1, 'snap': 1, 'transcripts': 1}) and generated the final annotation files, and then used of PASA[40] to update the EVM consensus predictions, added UTR annotations and models for alternatively spliced isoforms. The protein-coding genes which cannot hit to nr db by DIAMOND blastp (version v0.9.22.123)[45] with e value 1e-5 were removed.

To process small RNA data, we removed small RNA sequencing raw reads with Phred quality score less than 20, and adaptor sequences were trimmed. Processed reads of length 18–27 bp were then mapped to their respective horseshoe crab genome and analysed using the mirDeep2 package[46]. To identify conserved microRNAs, the predicted horseshoe crab microRNA hairpins were compared against metazoan microRNA precursor sequences from miRBase[28] using BLASTn (e value 0.01)[47]. Predicted microRNAs were manually examined. Novel microRNAs were defined only when they fulfilled the unique features of microRNAs (MirGeneDB 2.0 https://mirgenedb.org/information)[29]. In addition, the copy number of microRNA loci was examined by using microRNA hairpins confirmed above to BLAST against each horseshoe crab genome.

**Annotation of repetitive elements**. Repetitive elements were identified using an in-house pipeline. First, elements were identified using RepeatMasker ver. 4.0.8[48] with the *Arthropoda* RepBase[49] repeat library. Low-complexity repeats were ignored (-nolow) and a sensitive (-s) search was performed. Following this, a de novo repeat library was constructed using RepeatModeler ver. 1.0.11[50], including RECON ver. 1.08[51] and RepeatScout ver. 1.0.5[52]. Novel repeats identified by RepeatModeler were analysed with a 'BLAST, Extract, Extend' process to characterise elements along their entire length[53]. Consensus sequences and classification information for each repeat family were generated. The resulting de novo repeat library was utilised to identify repetitive elements using RepeatMasker. Repetitive element association with genomic features were determined using BedTools ver. 2.26.0[54]. 'Genic' repetitive elements were defined as those overlapping loci annotated as genes ± 2 kb and identified using the BedTools window function. All plots were generated using Rstudio ver. 1.2.1335 with R ver. 3.5.1[55] and ggplot2 ver. 3.2.1[56].

**Annotation of gene families and phylogenetic analyses**. Potential gene family sequences were first retrieved from the two genomes using tBLASTn[47]. Identity of each putatively identified gene was then tested by comparison to sequences in the NCBI nr database using BLASTx. For homeobox gene retrieval, sequences were also analysed using the BLAST function in HomeoDB. For phylogenetic analyses of gene families, DNA sequences were translated into amino acid sequences and aligned to other members of the gene family; gapped sites were removed from

alignments and phylogenetic trees were constructed using MEGA. Homeobox genes in the *Limulus* genome assembly (GCF_000517525) was obtained from previous study[8].

**Synteny analyses**. Synteny blocks were computed using SyMAP v4.2 (Synteny Mapping and Analysis Program) with default parameters except Min Dots from 2 to 7 (Minimum number of anchors required to define a syntenic block = 2–7) and 'mask_all_but_genes = 1' to mask non-genic sequence[57].

**Population genomic analyses**. After quality control using FastQC[32], adaptors and low-quality bases were removed from the read ends using FASTP[58] with '–qualified_quality_phred 30–length_required 25' and other default parameters, followed by a second round of quality control using FastQC. The trimmed reads were mapped to the unmasked mitochondrion genome (NC_012574 of *T. tridentatus* and NC_019623 of *C. rotundicauda*) using bwa (version 0.7.12-r1039) with default parameters. The mapped reads were sorted usning SortSam of picard, and duplicated reads were removed using MarkDuplicates of picard. Haplotype-Caller from the Genome Analysis Toolkit GATK (version 4, https://gatk.broadinstitute.org/hc/en-us) was used to estimate the general variant calling file for each individual, and then combined by GenotypeGVCFs to a single variant calling file. Hard filtering of the SNP calls was carried out with Fisher strand bias (FS > 60.0), mapping quality MQ < 40.0, and thresholding by sequencing coverage based on minimum coverage (DP < 100) and maximum coverage (DP > 1500). The SNPs were annotated with SnpEff (version 4.3T, http://snpeff.sourceforge.net/index.html)[59].

Filtered SNPs were used to generate population tree. The model-based software programme STRUCTURE Version 2.3.4. 81 was used for population analysis. To determine most appropriate *k* value, burn-in Markov Chain Monte Carlo (MCMC) replication was set to 50,000 and data were collected over 100,000 MCMC replications in each run. Two independent runs were performed setting the number of population (*k*) from 2 to 10 using a model allowing for admixture and correlated allele frequencies. The basis of this kind of clustering method is the allocation of individual samples to k clusters. The *k* value was determined based on the rate of change in LnP(D) between successive *k*, stability of grouping pattern across two run and sample information about the material in Supplementary Data 7, Supplementary Tables 9 and 10. Evolutionary divergence of within and between four different location horseshoe crab samples was performed using MEGA 7 (Molecular Evolutionary genetic analysis)[60] following maximum composite likelihood model with 1000 bootstrap iterations of all samples. Principal coordinate analysis (PCoA) and UPGMA phylogenetic analysis was conducted to further assess the population subdivisions. PCoA was performed based on distance matrix using DARwin V.6.0.21 and UPGMA tree was constructed based on the simple matching dissimilarity (DARwin).

Trimmed reads were mapped to the homeodomain sequences using bwa (version 0.7.12-r1039) with default parameters. The mapped reads were sorted using SortSam of picard, and duplicated reads were removed using MarkDuplicates of picard. HaplotypeCaller from the Genome Analysis Toolkit GATK (version 4, https://gatk.broadinstitute.org/hc/en-us) was used to estimate the general variant calling file for each individual, and then combined by GenotypeGVCFs to a single variant calling file. Hard filtering of the SNP calls was carried out with Fisher strand bias (FS > 60.0), mapping quality (MQ < 40.0), QualByDepth (QD < 2.0), MappingQualityRankSumTest (MQRankSum < −12.5), ReadPosRankSumTest (ReadPosRankSum < −8.0) as https://gatkforums.broadinstitute.org/gatk/discussion/2806/howto-apply-hard-filters-to-a-call-set. The filtered out SNPs were then annotated with SnpEff (version 4.3 T, http://snpeff.sourceforge.net/index.html)[59]. The missense mutation of the homeobox domain were manually checked with samtools tview.

**MicroRNA arm switching detection**. The expression levels of 5p and 3p arms of microRNAs in the horseshoe crabs were calculated based on the number of sequencing reads mapped to the respective arm region in the predicted microRNA hairpin using bowtie/mirDeep2. The expression of different arms of microRNAs from different species were mapped according to the previous method[61] or referred to the data from MirGeneDB 2.0[29]. The arm usage ratio (AUR) of each microRNA was calculated using the formula $AUR = 5p/(5p + 3p)$, where 5p and 3p refer to the read counts of predicted 5p and 3p arms, respectively. The AUR ranged from 0 to 1, with smaller values indicating the tendency of 3p preference and larger values indicating the tendency of 5p preference. 5p and 3p dominance was defined where AUR > 0.7 and <0.3, respectively. No arm preference was defined when AUR ranged from 0.3 to 0.7. The overall arm preference (OAP) of each horseshoe crab microRNA was defined by evaluating their arm dominance in multiple tissue samples. If more than 70% of all tissue samples showed one type of arm dominance, then this type of arm dominance was defined as the OAP of this microRNA. Otherwise, no OAP was defined.

**Statistics and reproducibility**. Sample size is outlined in Supplementary Data 7. All analyses are reproducible with access to genetic data (see 'Data availability').

**Reporting summary**. Further information on research design is available in the Nature Research Reporting Summary linked to this article.

## Data availability
The final genome assemblies have been deposited on NCBI with accession numbers WCHO00000000 and WCHN00000000, The raw reads generated in this study have been deposited to the NCBI database under the BioProject accession no. PRJNA574021 and PRJNA574023, for *C. rotundicauda* and *T. tridentatus* respectively. The genome annotation files are deposited in Figshare https://doi.org/10.6084/m9.figshare.13172414[62]. All other data, if any, are available upon reasonable request.

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

## Acknowledgements

The authors thank Peter Holland for discussion, B. Desany for helping with the assembly of 10X Genomics data, and F. Cheung, R. Leung, W. Tong, W. Yiu, and H. Yu for collection of some of the RNA. This research was supported by the Hong Kong Research Grant Council GRF Grant 14103516 and 14100919; Environment and Conservation Fund Project 28/2017; Agriculture, Fisheries and Conservation Department of HKSAR Government, and the School of Life Sciences of The Chinese University of Hong Kong (JHLH). P.Y.Q. and J.W.Q. are supported by Hong Kong Branch of Southern Marine Science and Engineering Guangdong Laboratory (Guangzhou) (SMSEGL20SC01). A.H. is supported by a Biotechnology and Biological Sciences Research Council (BBSRC) David Phillips Fellowship (BB/N020146/1). T.B. is supported by a studentship from the Biotechnology and Biological Sciences Research Council-funded South West Biosciences Doctoral Training Partnership (BB/M009122/1).

## Author contributions

J.H.L.H. conceived and supervised the study. W.N. carried out the genome assemblies and analyses, gene model predictions, microRNA mapping, and synteny analyses. Z.Q. carried out the microRNA annotation, final checking microRNA copies and arm switching analyses. Y.L. carried out the homeobox gene analyses, synteny analyses, and the SNP analyses in populations. T.B.O. carried out the homeobox gene identifications and tree construction. A.Y.P.W. carried out the gene and microRNA copies analyses and arm switching analyses. H.Y.Y. provided animal husbandry and logistics. H.T.L. carried out novel microRNA analyses. S.N. carried out the population structure analyses. T.B. and A.H. performed the T.E. analyses. T.S. involved in the final version of genome assembly, and J.C. involved in earlier version of genome assembly. T.F.C., H.S.K., S.M.N., G.P., P.Y.Q., J.W.Q., K.Y.Y., S.S.T., W.G.B., S.G.C., J.H.L.H. applied and obtained the funding. N.I., S.P., A.J., S.G.C. collected and provided samples in field. S.S.T., W.G.C., S.G.C., A.H., J.H.L.H. drafted the first version of the manuscript. All authors provided comments and approved the manuscript.

## Competing interests

The authors declare no competing interests.

## Additional information

[1]School of Life Sciences, Simon F.S. Li Marine Science Laboratory, State Key Laboratory of Agrobiotechnology, The Chinese University of Hong Kong, Hong Kong, China. [2]Centre for Ecology and Conservation, University of Exeter, Penryn, UK. [3]Dovetail Genomics, Scotts Valley, CA, USA. [4]State Key Laboratory of Agrobiotechnology, School of Life Sciences, The Chinese University of Hong Kong, Hong Kong, China. [5]School of Life Sciences, The Chinese University of Hong Kong, Hong Kong, China. [6]School of Biological Sciences, The University of Hong Kong, Hong Kong, China. [7]Leibniz Institute of Natural Product Research and Infection Biology – Hans Knöll Institute, Jena, Germany. [8]Department of Ocean Science and Hong Kong Branch of Southern Marine Science and Engineering Guangdong Laboratory (Guangzhou), Hong Kong University of Science and Technology, Hong Kong, China. [9]Department of Biology, Hong Kong Baptist University, Hong Kong, China. [10]Department of Computer Science and Engineering, The Chinese University of Hong Kong, Hong Kong, China. [11]Institute of Marine Biotechnology, Universiti Malaysia Terengganu, Terengganu, Malaysia. [12]Department of Bioscience and Biotechnology, Fakir Mohan University, Balasore, India. [13]Institute of Tropical Biodiversity and Sustainable Development, University Malaysia Terengganu, 20130 Kuala Nerus, Terengganu, Malaysia. [14]Research Division, Association for Biodiversity Conservation and Research (ABC), Odisha 756003, India. [15]Institute of Oceanography and Maritime Studies (INOCEM), Kulliyyah of Science, International Islamic University, Kuantan, Malaysia. [16]Department of Cell and Systems Biology, University of Toronto, Toronto, Canada. [17]Department of Biology, Queen's University, Toronto, Canada. [18]Department of Chemistry, City University of Hong Kong, Hong Kong, China. [19]These authors contributed equally: Wenyan Nong, Zhe Qu, Yiqian Li, Tom Barton-Owen, Annette Y. P. Wong. ✉email: jeromehui@cuhk.edu.hk

