## [Peer Review File · Communications Biology]

Reviewers' Comments:

Reviewer #1:

Remarks to the Author:

The Hui lab presents yet another very interesting paper this year and I am very curious about the topic. The manuscript "Horseshoe crab genomes reveal the evolutionary fates of genes and microRNAs after 2 three rounds (3R) of whole genome duplication" is interesting and important. I have several comments on it and would like to start with one comment that is regarding a trend in the Hui labs papers that I find a little discomforting and in my opinion highly unusual: Including the presented manuscript, 3 papers were released on biorxiv, and one of them published with MBE now, that all have 5 shared first authors (1,2). In order to maintain some level of transparency here, at least for me if not demanded by the journal, I would like to ask for a detailed statement about the contributions of these 5 persons to the submitted manuscript.

WGD-duplication: Have the authors attempted to make a selection analysis on the miRNA genes, i.e. can they compare the conservation of genes between and within species with few or many paralogues?

For identical paralogues. Have the loci been validated by gPCR - can the authors be sure that these are actual copies and not misassemblies?

smallRNA read data: it is unclear how many tissues of each species in how many replicates (if any!) were sequenced. The authors should make this clear and could use miRTrace (3) for their analysis to make it more transparent and make a nice SI-file being the html report file for each species.

Novel microRNAs. miRDeep2 is known to produce high false-positives and it is therefore nice to read that the authors follow Fromm et al 2015 in the annotation criteria of novel microRNAs. Could they please provide structures and read distribution (pdfs from miRDeep?).

Submission to miRBase: The PI is a former lab member of the Griffiths-Jones lab. Could you make sure that these new sequences are actually going to be released in miRBase? There is a long backlog for miRBase' submissions and if the authors cannot guarantee that miRBase would release their data within the next 3 months, they should put them for instance on their lab-website.

Figures. In general I think the figures are below the expected quality. They use too many colors (Fig 1 & 2), confusing heatmap scaling (Fig 2 & 4) and confusing & arbitrary selection of examples (Fig 4 & 5)

REFERENCES

1. Qu Z, Nong W, So WL, Barton-Owen T, Li Y, Li C. Millipede genomes reveal unique adaptation of genes and microRNAs during myriapod evolution. bioRxiv [Internet]. biorxiv.org; 2020; Available from: <https://www.biorxiv.org/content/10.1101/2020.01.09.900019v1.abstract>
2. Qu Z, Yiu WC, Yip HY, Nong W, Yu CWC, Lee IHT, Wong AYP, Wong NWY, Cheung FKM, Chan TF, et al. MicroRNA clusters integrate evolutionary constraints on expression and target affinities: the miR-6/5/4/286/3/309 cluster in Drosophila. Mol Biol Evol [Internet]. 2020; Available from: <http://dx.doi.org/10.1093/molbev/msaa146>
3. Kang W, Eldfjell Y, Fromm B, Estivill X, Biryukova I, Friedländer MR. miRTrace reveals the organismal origins of microRNA sequencing data. Genome Biol. 2018;19:213.

Reviewer #2:

Remarks to the Author:

In this manuscript, Nong et al report the genomes of two horseshoe crab species, *Carcinoscorpius rotundicauda* and *Tachypleus tridentatus*. Horseshoe crabs are one of the most evolutionary relevant arthropod lineages and are of fundamental importance for understanding the evolution of chelicerates. They are also among the few described non-vertebrate lineages that have undergone whole genome duplications (WGDs) and are therefore of great interest to investigate the evolutionary consequences of these large-scale mutational events. Furthermore the immune system of horseshoe crabs is of special relevance and the haemolymph of these species has been widely used in biomedical applications (with the consequent negative impact in horseshoe crab populations).

The genomes of horseshoe crabs have received quite a lot of attention in the last years, and 5 different genomic assemblies with different degrees of completeness and contiguity have been published for members of the 3 extant horseshoe crab genera. However, no systematic study comparing genomic data across the 3 horseshoe crab genera has been done yet, and many questions remain open, in particular regarding the number of rounds of WGDs present in their genomes.

The study of Nong et al does precisely that and provides a detailed comparison between *Carcinoscorpius*, *Tachypleus* and *Limulus*, characterizing in detail a superfamily of transcription factors, the homeobox genes, and miRNA complements. Thus, this manuscript will become a reference work in the field as the first taxonomically comprehensive study of horseshoe crab genomes and their WGDs, providing a global picture for horseshoe crab genomics. They clearly establish that this lineage underwent 3 rounds of WGDs and show the different fates that the duplicated genes have followed in the different species and populations.

Overall, the manuscript is well written and presented.

I only have two comments (related to the lack of support of a shared WGD with scorpions and spiders and to improving the naming and orthology assignment of Hox clusters) and a list of several very minor issues:

1. Shared WGD. The authors try to address if the first and most ancient round of horseshoe crab WGDs is shared or not with the ancient WGD present in arachnospulmonates (i.e. spiders and scorpions). To address this, they (i) performed analyses of conserved synteny and, (ii) reconsidered recent evidence on phylogenetic relationships within the Chelicerata, concluding that the WGD is shared and ancestral to the two chelicerate lineages. This is surprising given that all previously published data (phylogenetic trees for several duplicated gene families), including results published by some of the authors of the current manuscript (Kenny et al 2016, Schwager et al 2017, Shingate et al 2020) support the opposite scenario, finding that most trees support in-paralogous relationships and a general lack of support for a shared WGD, therefore favouring that the WGDs occurred in the two chelicerate lineages independently (one round in the last common ancestor of spiders and scorpions and three rounds in horseshoe crabs). I agree with the authors that these results need further support and that sometimes phylogenetic trees are not able to solve if ancient WGDs are ancestral or independent between different lineages (such as in cases of 'tetralogy', in lineages that have diverged very rapidly just after the WGD events took place, when re-diploidization may have not been completed and homeologous recombination may still be happening (see Martin & Holland MBE 2014). However, contrary to the author's interpretations, I do not think that the analyses presented here add support to the shared WGD scenario that could justify disregarding the previous tree-based evidence.

First, that the authors found conserved syntenic blocks around the Hox clusters of horseshoe crabs, spiders and scorpions only indicates that there is conserved macro (or micro) synteny in these genomic regions between these lineages, but this conservation could be equally found regardless if the WGD are shared or not. Conservation of synteny around Hox genes have been found across different lineages, and especially between vertebrates, which have 2 rounds of WGDs and amphioxus, whose genome is not duplicated (see for instance Acemel et al 2016). Thus, I do not see how the circos plots presented in fig 3B suggest that the duplications are shared.

Second, regarding the phylogenetic position of horseshoe crabs, I agree that a close phylogenetic relationship between xiphosuran and arachnospulmonates would make the shared WGD less unlikely, opening the possibility for a 'tetralogy'-like scenario. However, this tree topology is not enough by itself to support that the WGD is ancestral, especially taking into account that chelicerate phylogenetic relationships are far from being solved. In fact, if the alternative topology where arachnids are monophyletic is correct (see Lozano-Fernandez et al 2019), this would completely rule out the possibility of a shared WGD: the genome structure of mites and ticks (with a single Hox cluster and not a single piece of evidence of paralogous chromosomal segments) cannot be the result of an ancestral WGD followed by extensive gene loss as suggested in lines 231-234.

Taking all this into account, I would suggest that the authors follow one of these two options:

- Changing a bit the writing in some sections of the paper, including: 1. rewriting the synteny section as a section on conserved synteny around Hox genes but without claiming that these results support a shared WGD, 2. Eliminate the corresponding claim in the abstract and other parts of the text and 3. Include the arguments related to chelicerate phylogeny only as discussion, stating that depending on which is the correct phylogenetic position of horseshoe crabs, additional analyses and data may be required to fully demonstrate that the WGDs are not shared if xiphosuran and arachnospulmonates are in fact closely related.

- Perform thorough additional analyses that convincingly show that due to 'tetralogy' or whatever other reason, all previously published gene family trees suggesting independent WGD events were not recovering the real topology and evolutionary events. For that, I think the only possibility would be that the authors are able to find cases of rare genomic changes that are synapomorphic for xiphosuran and arachnospulmonates and are therefore not present in the outgroups, including other chelicerate lineages such as mites and ticks, which in this scenario would also be outgroups. Then, it would be necessary to find that some of these synapomorphic rare genomic changes are present in several ohnologous copies (at least twice) in the genomes of both horseshoe crabs and at least one of the arachnospulmonate lineages. These rare genomic changes could include novel microsyntenic associations (see for instance the novel linkage between Hedgehog and Lmbr1 genes in vertebrates, which is present twice in vertebrate genomes, Shh and Lmbr1 and Dhh and Lmbr1l, see Royo et al Sci. Rep 2012), but also intron gains that are shared by several ohnologues, ohnologous retrogenes, ohnologous chimeric genes or with novel domain combinations...

Please note that I am not asking the authors to perform these extra analyses. If the authors decide just to change the text and eliminate the claim that the duplication is shared, that would be absolutely fine. The analyses would only be required if the authors decide to keep this statement.

2. Hox clusters and other homeobox genes. This is one of the nicest parts of the manuscript since it shows for the first time together data from the 3 horseshoe genera, clearly showing that these species have a pattern of duplications consistent with 3 rounds of WGDs. Importantly, the Hox analyses presented here are much more comprehensive than previous studies and the authors have managed to identify many more Hox genes than them. Furthermore, this is the first time that clusters from all extant xiphosuran genera are compared and shown together. However, I have some comments regarding the way these data are presented. Given that this manuscript is going to be the reference work in that regard, I think it would be good to establish a consensus nomenclature for horseshoe crabs Hox clusters duplicates. Thus, I recommend that the authors follow a similar system to that used in vertebrates, as it has recently been done in Fig.3 in Shingate et al 2020. So, it would be great that they compare their data with those of Shingate et al and name the Hox clusters accordingly, that is, clusters HoxA, HoxB, HoxC, HoxD and HoxE. Similarly, I think it is better to display the different clusters in different lines, even in cases where two clusters are found in the same chromosome. The fact that this linkage exists can be perfectly indicated with the scaffold number, and, if necessary, it could be further emphasized in the figure caption and/or the main text. Furthermore, since the two clusters are really far away from each other, this is probably the result of a chromosome fusion and many readers maybe misled by this and think that this is a tandem duplication. Similarly, the linkage between Hox and GsxA distracts

from the main message in the figure, so I think it could be indicated in a less prominent way (in the caption, the main text or in a suppl. figure specifically done for homeobox linkages). Furthermore, it would be better if Hox paralogous groups are all shown aligned by columns, so readers can immediately spot which genes have been lost in each cluster (again following a similar outline as the one used by Shingate et al). I think these changes are quite important because they will make the entire figure much easier to interpret, so one can immediately see how many Hox clusters are present in horseshoe crabs. And these changes should be applied to the 3 horseshoe crab genera, naming orthologous clusters in a consistent manner (perhaps in some cases this will require performing gene trees of specific paralogous groups to identify orthologs across the 3 species). Finally, whenever possible (I understand this could be tricky in some cases), the authors could also try to name all the Hox genes found in isolated scaffolds in a consistent manner across the 3 species and assign them to now dispersed ancestral clusters, taking advantage of the linkage and synteny information found in the different assemblies and species and/or results from trees. The same improvements could be applied to other homeobox families, but I understand that this would imply a lot of work and it is beyond the scope of this manuscript, so I do not think it is necessary to do these analyses with the rest of homeobox genes (other ANTP, PRD, TALE etc). But, as stated by the authors, Hox clusters are iconic for WGDs and it is important that the results related to them are sufficiently clear and detailed.

3. Minor comments

A. As a general comment, the resolution of the figures is very low, so most of the text included within the figures is impossible to read (scaffolds numbers etc.). I have checked the figures and tried enlarged them by zooming in both in the merged pdf and in the original doc file provided by the authors, but in both of them the resolution was equally low.

B. I understand the paper by Shingate et al NatComm 2020 was probably published after Nong et al submitted the current manuscript, but now that it has been published I think it is important that the authors reference this work as they did with other horseshoe crab genome papers. Similarly, in addition to Nossa et al, it is also important to cite the other publication reporting a *Limulus* genome assembly: Battelle et al GBE 8, 1571–1589 (2016), especially taking into account that the assembly reported by Battelle et al is by far the best publicly available *Limulus* assembly.

C. The methods do not include a section describing how the authors searched for homeobox genes in *Limulus* and which *Limulus* assembly version was used (perhaps the one by Battelle et al GCF_000517525.1 ??). I think it is important to detail this information.

D. Table in Fig1. Since Nong et al manuscript compares information from all available xiphosuran genomes and will be a key reference in this regard, I think it would be good to add to this table the stats and accession numbers of the other *Carcinoscorpius* assembly (the one by Shingate et al) and the *Limulus* assembly from Battelle et al that I mentioned in my previous comments.

E. In line 113 the authors say

“Furthermore, 89.8% of the sequences assembled for *C. rotundicauda* genome are contained on just 16 pseudomolecules, consistent with a near chromosome-level assembly (chromosome $2n=32$, Iwasaki et al 1988, Supplementary information S1, Table 1.1.3)”

Could the authors provide the equivalent information about pseudomolecules and karyotype of the other genome assembly reported in the manuscript, *T. tridentatus*?

F. In line 167 the authors comment on the different number of Hox clusters found by Gong et al:

“In a recent study of the *T. tridentatus* re-sequenced genome, the authors could only find two Hox clusters and could not identify the *Ftz* gene inside these clusters (Gong et al 2019). On contrary, our results suggested that there are three Hox clusters (including *Ftz*), and thus more than one round of WGD occurred in the lineage leading to extant horseshoe crabs.”

Could the authors elaborate further on these differences? Are these genes missing in Gong et al's assembly because of incompleteness? Or are these genes present in the assembly but they were missed by Gong et al because they are not annotated? Have the authors tried to search for these genes in Gong et al's assembly using blast? I think it would be important for some readers to understand the source of these discrepancies. The same applies to *Carcinoscorpius*, it would be good if the authors could clarify why they found more Hox genes that were missed in Shingate et al paper.

G. Supplementary figure with miRNAs alignments. In some cases only the scaffold numbers are indicated and the abbreviation used for each of the species is missing. Please, revise.

H. I found quite interesting that 5 of the novel xiphosuran miRNAs families have several duplicates, these are Novel 2, Novel 3, Novel 4, Novel 5 and especially Novel 10 (which has 4 copies). Could these duplicated miRNAs be considered as duplicated rare genomic changes that support an independent origin of the horseshoe crab WGDs, as I discussed above in my first comment? Are these duplicates found in tandem or in ohnologous regions? It would be interesting that the authors discuss this in the manuscript.

I. In line 294 the authors state:

"As these genomes have undergone WGD, to confidently reveal their population structure, we only mapped sequencing reads to the mitochondrial genome and constructed the evolutionary trees from mitochondrial data."

I do not see the relationship between the presence of ancient WGDs and the impossibility of studying populations using nuclear data, if that were really the case, hundreds of population studies performed in vertebrates using nuclear data could not exist. I do not have any problem with the authors' decision to focus on mitochondrial data, but this statement does not make sense. I suggest to delete it or to use a more convincing argument.

Reviewers' comments:

Reviewer #1 (Remarks to the Author):

1. The Hui lab presents yet another very interesting paper this year and I am very curious about the topic. The manuscript “Horseshoe crab genomes reveal the evolutionary fates of genes and microRNAs after 2 three rounds (3R) of whole genome duplication” is interesting and important. I have several comments on it and would like to start with one comment that is regarding a trend in the Hui labs papers that I find a little discomforting and in my opinion highly unusual: Including the presented manuscript, 3 papers were released on biorxiv, and one of them published with MBE now, that all have 5 shared first authors (1,2). In order to maintain some level of transparency here, at least for me if not demanded by the journal, I would like to ask for a detailed statement about the contributions of these 5 persons to the submitted manuscript.

Response 1:

We thank the reviewer for their positive comments regarding our manuscripts. We appreciate their concerns, and agree thoroughly that it is extremely important to be open and transparent about authorships. Consequently, we are very happy to provide further details of the background behind our authorships. Firstly, we would like to reassure the reviewer that we always assign co-first authorships, where deemed relevant, in accordance with equal contributions. Secondly, we consider that this issue arises given that large genome projects are often put together in a very different way, compared to the more traditional model of a study being led by a single first author who writes the whole manuscript. In our case, the five shared authors all have respective areas of expertise, and all contributed written sections. Specifically, WN carried out the genome assemblies and analyses, gene model predictions, microRNA mapping, and synteny analyses. ZQ carried out the microRNA annotation, final checking microRNA copies and arm switching analyses. YL carried out the homeobox gene analyses, synteny analyses, and the SNP analyses in populations. TBO carried out the homeobox gene identifications and tree construction. AYPW carried out the gene and microRNA copies analyses and arm switching analyses. All five first authors contributed to the final manuscript. We are very happy for these details to be included in our manuscript if necessary.

2. WGD-duplication: Have the authors attempted to make a selection analysis on the miRNA genes, i.e. can they compare the conservation of genes between and within species with few or many paralogues?

Response 2:

We thank the reviewer for this suggestion. The conservation of microRNAs within the same species, and between different species, has now been compared and this information is provided in Supplementary information S1 Figure S1.2.12.

3. For identical paralogues. Have the loci been validated by gPCR - can the authors be sure that these are actual copies and not misassemblies?

Response 3:

We agree that this is also good suggestion. A total of 20 and 6 microRNA paralogues were identified to possess identical sequences in the horseshoe crabs *Carcinoscorpius rotundicauda* (Cr) and *Tachypleus tridentatus* (Tt), respectively. We carried out gPCR validation for two of these microRNAs (mir-5305 and mir-87) with primers designed on the flanking sequences. As shown in the gel picture below, our results confirmed the presence of these identical paralogues.

Gel picture showing the gPCR validation results. Lane 1-5: gPCR of Tt; Lane 1-2: 2 paralogues of mir-87; Lane 3-4: 2 paralogues of mir-5305, Lane 5: COI as positive control; Lane 6: DNA marker; Lane 7-16: gPCR of 2 individuals of Cr. Lane 7-10: 2 paralogues of mir-87; Lane 11-14: 2 paralogues of mir-5305; Lane 15-16: COI of 2 individuals of Cr.

If the reviewer still has concerns, we have also counted identical microRNA paralogues to test if the 3R hypothesis should be rejected. As one can see below, the hypothesis still holds.

4. smallRNA read data: it is unclear how many tissues of each species in how many replicates (if any!) were sequenced. The authors should make this clear and could use miRTrace (3) for their analysis to make it more transparent and make a nice SI-file being the html report file for each species.

Response 4:

A total of 7 and 21 small RNA samples (n =1) have been sequenced for *C. rotundicauda* and *T. tridentatu* respectively. The information is included in Supplementary information S1 Table 1.1.5.and Table 1.1.6. We are confident of our claims, given that the cases of microRNA arm switching are found in multiple samples, and so form natural replicates for one another. We agree that it is important to provide all the requested information, and have adjusted our manuscript in accordance. Please see below.

Table 1.1.5. *C. rotundicauda* transcriptome sequencing data information

Type	Stages	Platform	Reads	Bases	Accession
mRNA	Blood	NovaSeq 6000	38,863,153	5,829,449,781	SAMN13738014
	Brain		31,575,827	4,736,353,472	SAMN13738015
	chelicerate		39,549,495	5,932,400,015	SAMN13738016
	Heart		27,783,626	4,167,527,190	SAMN13738017
	Leg1		35,135,528	5,270,307,850	SAMN13738018
	Leg5		40,202,979	6,030,422,836	SAMN13738019
	Tail		33,116,985	4,967,527,398	SAMN13738020
Small RNA	Brain	NovaSeq 50SE	20,410,408	1,020,520,400	SAMN13871197
	Blood		19,564,687	978,234,350	SAMN13871198
	Chelicerae		19,393,640	969,682,000	SAMN13871199
	Heart		20,309,098	1,015,454,900	SAMN13871200
	1st_pair_of_leg		19,043,622	952,181,100	SAMN13871201
	5th_pair_of_leg		29,345,855	1,467,292,750	SAMN13871202
	Telson		22,375,032	1,118,751,600	SAMN13871203

Table 1.1.6. *T. tridentatus* transcriptome sequencing data information

Type	Stages	Platform	Reads	Bases	Accession
mRNA	fbl.blood.	NovaSeq 6000	37,977,195	5,696,301,567	SAMN13738021
	fbr.brain.		34,292,198	5,143,570,785	SAMN13738022
	fc.chelicerate.		32,805,893	4,920,650,285	SAMN13738023
	fh.heart.		35,230,816	5,284,362,030	SAMN13738024
	f11.leg1.		36,576,210	5,486,164,882	SAMN13738025
	f15.leg5.		33,927,286	5,088,844,698	SAMN13738026
	ft.telson.		39,699,452	5,954,617,801	SAMN13738027
	HSC.1st.Juv.		21,193,857	2,118,740,674	SAMN13738028
	HSC.2nd.Juv.		20,816,258	2,080,987,978	SAMN13738029
	HSC.3rd.Juv.		20,288,354	2,028,223,006	SAMN13738030
Small RNA	Adult_1st_pair_of_leg	HiSeq 2500	25,650,608	563,162,745	SAMN13871219
	Adult_5th_pair_of_leg		24,554,405	539,154,086	SAMN13871220
	Adult_blood		27,669,258	615,864,084	SAMN13871208
	Adult_brain		26,993,521	617,921,430	SAMN13871210
	Adult_chelicerae		31,948,259	703,790,441	SAMN13871211
	Adult_heart		26,082,982	575,132,790	SAMN13871217
	Adult_telson		26,857,159	605,349,870	SAMN13871223
	Juvenile_1st_pair_of_l eg		14,965,599	332,545,166	SAMN13871221
	Juvenile_5th_pair_of_l eg		17,705,657	387,406,260	SAMN13871222
	Juvenile_blood		17,054,020	405,397,024	SAMN13871209

	Juvenile_chelicerae		14,910,286	327,534,454	SAMN13871212
	Juvenile_heart		13,563,640	310,098,352	SAMN13871218
	Juvenile_telson		12,047,257	265,525,861	SAMN13871224
	1st_instar	NovaSeq 50SE	27,947,739	611,436,838	SAMN13871204
	2nd_instar		24,780,645	557,062,135	SAMN13871205
	3rd_instar		25,705,620	558,368,223	SAMN13871206
	Adult_blood		25,362,968	559,120,297	SAMN13871207
	Adult_heart		20,381,064	452,114,559	SAMN13871216
	Egg_1		20,641,958	494,478,133	SAMN13871213
	Egg_2		22,755,433	549,370,934	SAMN13871214
	Gonad		24,754,354	578,404,342	SAMN13871215

Furthermore, miRTrace outputs for the two horseshoe crabs are now included in supplementary information S1 under Table 1.1.6, shown as follows:

C. rotundicauda: http://137.189.43.6/miRTrace_output/cro/mirtrace-report.html

T. tridentatus: http://137.189.43.6/miRTrace_output/ttr/mirtrace-report.html

5. Novel microRNAs. miRDeep2 is known to produce high false-positives and it is therefore nice to read that the authors follow Fromm et al 2015 in the annotation criteria of novel microRNAs. Could they please provide structures and read distribution (pdfs from miRDeep?).

Response 5:

The miRDeep2 output files are now provided in Supplementary files S5 and S6.

6. Submission to miRBase: The PI is a former lab member of the Griffiths-Jones lab. Could you make sure that these new sequences are actually going to be released in miRBase? There is a long backlog for miRBase' submissions and if the authors cannot guarantee that miRBase would release their data within the next 3 months, they should put them for instance on their lab-website.

Response 6:

The raw reads have been submitted to NCBI with accession numbers as shown in Table 1.1.5 and Table 1.1.6.

Regarding the submission of miRNAs to miRBase, this has also been carried out as suggested.

7. Figures. In general I think the figures are below the expected quality. They use too many colors (Fig 1 &2), confusing heatmap scaling (Fig 2 &4) and confusing & arbitrary selection of examples (Fig 4 &5)

Response 7:

The figures have been adjusted in accordance with the reviewers suggestions. Basically only key message remains in the main figures, while the other information is now included in the supplementary information.

Figure 1.

Figure 3

Figure 4.

Figure 5.

Reviewer #2 (Remarks to the Author):

8. In this manuscript, Nong et al report the genomes of two horseshoe crab species, *Carcinoscorpius rotundicauda* and *Tachypleus tridentatus*. Horseshoe crabs are one of the most evolutionary relevant arthropod lineages and are of fundamental importance for understanding the evolution of chelicerates. They are also among the few described non-vertebrate lineages that have underwent whole genome duplications (WGDs) and are therefore of great interest to investigate the evolutionary consequences of these large-scale mutational events. Furthermore the immune system of horseshoe crabs is of special relevance and the haemolymph of these species has been widely used in biomedical applications (with the consequent negative impact in horseshoe crab populations).

The genomes of horseshoe crabs have received quite a lot of attention in the last years, and 5 different genomic assemblies with different degrees of completeness and contiguity have been published for members of the 3 extant horseshoe crab genera. However, no systematic study comparing genomic data across the 3 horseshoe crab genera has been done yet, and many questions remain open, in particular regarding the number of rounds of WGDs present in their genomes.

The study of Nong et al does precisely that and provides a detailed comparison between *Carcinoscorpius*, *Tachypleus* and *Limulus*, characterizing in detail a superfamily of transcription factors, the homeobox genes, and miRNA complements. Thus, this manuscript will become a reference work in the field as the first taxonomically comprehensive study of horseshoe crab genomes and their WGDs, providing a global picture for horseshoe crab genomics. They clearly establish that this lineage underwent 3 rounds of WGDs and show the different fates that the duplicated genes have followed in the different species and populations.

Overall, the manuscript is well written and presented.

I only have two comments (related to the lack of support of a shared WGD with scorpions and spiders and to improving the naming and orthology assignment of Hox clusters) and a list of several very minor issues:

1. Shared WGD. The authors try to address if the first and most ancient round of horseshoe crab WGDs is shared or not with the ancient WGD present in arachnoplumonates (i.e. spiders and scorpions). To address this, they (i) performed analyses of conserved synteny and, (ii) reconsidered recent evidence on phylogenetic relationships within the Chelicerata, concluding that the WGD is shared and ancestral to the two chelicerate lineages. This is surprising given that all previously published data (phylogenetic trees for several duplicated gene families), including results published by some of the authors of the current manuscript (Kenny et al 2016, Schwager et al 2017, Shingate et al 2020) support the opposite scenario, finding that most trees support in-paralogous relationships and a general lack of support for a shared WGD, therefore favouring that the WGDs occurred in the two chelicerate lineages independently (one round in the last common ancestor of spiders and scorpions and three rounds in horseshoe crabs). I agree with the authors that these results need further support and that sometimes phylogenetic trees are not able to solve if ancient

WGDs are ancestral or independent between different lineages (such as in cases of ‘tetralogy’, in lineages that have diverged very rapidly just after the WGD events took place, when re-diploidization may have not been completed and homeologous recombination may still be happening (see Martin & Holland MBE 2014). However, contrary to the author’s interpretations, I do not think that the analyses presented here add support to the shared WGD scenario that could justify disregarding the previous tree-based evidence.

First, that the authors found conserved syntenic blocks around the Hox clusters of horseshoe crabs, spiders and scorpions only indicates that there is conserved macro (or micro) synteny in these genomic regions between these lineages, but this conservation could be equally found regardless if the WGD are shared or not. Conservation of synteny around Hox genes have been found across different lineages, and especially between vertebrates, which have 2 rounds of WGDs and amphioxus, whose genome is not duplicated (see for instance Acemel et al 2016). Thus, I do not see how the circos plots presented in fig 3B suggest that the duplications are shared.

Second, regarding the phylogenetic position of horseshoe crabs, I agree that a close phylogenetic relationship between xiphosuran and arachnospulmonates would make the shared WGD less unlikely, opening the possibility for a ‘tetralogy’-like scenario. However, this tree topology is not enough by itself to support that the WGD is ancestral, especially taking into account that chelicerate phylogenetic relationships are far from being solved. In fact, if the alternative topology where arachnids are monophyletic is correct (see Lozano-Fernandez et al 2019), this would completely rule out the possibility of a shared WGD: the genome structure of mites and ticks (with a single Hox cluster and not a single piece of evidence of paralogous chromosomal segments) cannot be the result of an ancestral WGD followed by extensive gene loss as suggested in lines 231-234.

Taking all this into account, I would suggest that the authors follow one of these two options:

- Changing a bit the writing in some sections of the paper, including: 1. rewriting the synteny section as a section on conserved synteny around Hox genes but without claiming that these results support a shared WGD, 2. Eliminate the corresponding claim in the abstract and other parts of the text and 3. Include the arguments related to chelicerate phylogeny only as discussion, stating that depending on which is the correct phylogenetic position of horseshoe crabs, additional analyses and data may be required to fully demonstrate that the WGDs are not shared if xiphosuran and arachnospulmonates are in fact closely related.

- Perform thorough additional analyses that convincingly show that due to ‘tetralogy’ or whatever other reason, all previously published gene family trees suggesting independent WGD events were not recovering the real topology and evolutionary events. For that, I think the only possibility would be that the authors are able to find cases of rare genomic changes that are synapomorphic for xiphosuran and arachnospulmonates and are therefore not present in the outgroups, including other chelicerate lineages such as mites and ticks, which in this scenario would also be outgroups. Then, it would be necessary to find that some of these synapomorphic rare

genomic changes are present in several ohnologous copies (at least twice) in the genomes of both horseshoe crabs and at least one of the arachnoplumonate lineages. These rare genomic changes could include novel microsyntenic associations (see for instance the novel linkage between Hedgehog and Lmbr1 genes in vertebrates, which is present twice in vertebrate genomes, Shh and Lmbr1 and Dhh and Lmbr1, see Royo et al Sci. Rep 2012), but also intron gains that are shared by several ohnologues, ohnologous retrogenes, ohnologous chimeric genes or with novel domain combinations...

Please note that I am not asking the authors to perform these extra analyses. If the authors decide just to change the text and eliminate the claim that the duplication is shared, that would be absolutely fine. The analyses would only be required if the authors decide to keep this statement.

Response 8:

We thank the reviewer for all their detailed comments and suggestions. We have now modified our manuscript sections as suggested. The changes are summarised as follows:

a) synteny section – the original sentence on Page 8, lines 1-2, “Our data, suggested for the first time, that the WGD in horseshoe crab is a shared event with the WGD in spider and scorpion.” has been removed from the main text.

b) abstract – the original sentence on Page 2, lines 9-11, “Analyses of gene families, microRNAs, and synteny show that horseshoe crabs have undergone three rounds (3R) of WGD, and that these WGD events are shared with spiders.” has now been changed to “Analyses of gene families, microRNAs, and synteny show that horseshoe crabs have undergone three rounds (3R) of WGD.”

c) arguments to chelicerate phylogeny – the section is rewritten as suggested (Page 8, lines 11-14, and shown as follows “Consequently, additional analyses and data are needed to differentiate between these scenarios, and fully demonstrate whether a shared WGD event occurred at the common ancestor of horseshoe crabs, spiders, and scorpions, or if an ancestral WGD occurred in the ancestral lineage to chelicerates and xiphosurans, followed by massive gene losses in some lineages, such as ticks and mites.”

9. 2. Hox clusters and other homeobox genes. This is one of the nicest parts of the manuscript since it shows for the first time together data from the 3 horseshoe genera, clearly showing that these species have a pattern of duplications consistent with 3 rounds of WGDs. Importantly, the Hox analyses presented here are much more comprehensive than previous studies and the authors have managed to identify many more Hox genes than them. Furthermore, this is the first time that clusters from all extant xiphosuran genera are compared and shown together. However, I have some comments regarding the way these data are presented. Given that this manuscript is going to be the reference work in that regard, I think it would be good to establish a consensus nomenclature for horseshoe crabs Hox clusters duplicates. Thus, I recommend that the authors follow a similar system to that used in vertebrates, as it has recently been done in Fig.3 in Shingate et al 2020. So, it would be great that they

compare their data with those of Shingate et al and name the Hox clusters accordingly, that is, clusters HoxA, HoxB, HoxC, HoxD and HoxE. Similarly, I think it is better to display the different clusters in different lines, even in cases where two clusters are found in the same chromosome. The fact that this linkage exists can be perfectly indicated with the scaffold number, and, if necessary, it could be further emphasized in the figure caption and/or the main text. Furthermore, since the two clusters are really far away from each other, this is probably the result of a chromosome fusion and many readers maybe misled by this and think that this a tandem duplication. Similarly, the linkage between Hox and GsxA distracts from the main message in the figure, so I think it could be indicated in a less prominent way (in the caption, the main text or in a suppl. figure specifically done for homeobox linkages). Furthermore, it would be better if Hox paralogous groups are all shown aligned by columns, so readers can immediately spot which genes have been lost in each cluster (again following a similar outline as the one used by Shingate et al). I think these changes are quite important because they will make the entire figure much easier to interpret, so one can immediately see how many Hox clusters are present in horseshoe crabs. And these changes should be applied to the 3 horseshoe crab genera, naming orthologous clusters in a consistent manner (perhaps in some cases this will require performing gene trees of specific paralogous groups to identify orthologs across the 3 species). Finally, whenever possible (I understand this could be tricky in some cases), the authors could also try to name all the Hox genes found in isolated scaffolds in a consistent manner across the 3 species and assign them to now dispersed ancestral clusters, taking advantage of the linkage and synteny information found in the different assemblies and species and/or results from trees.

Response 9:

We agree with the reviewer regarding these important points, and have now taken the consensus nomenclature (HoxA, HoxB, HoxC, HoxD and HoxE) according to Shingate et al's paper. Furthermore, additional syntenic analyses were also carried out in order to confirm assignments.

	Cr NC (nccr)		Cr our study (hccr)	Tt our study(hctt)
HoxB cluster	Chr13	CM022403.1	Sc28yqQ_9435	Scaffold_44553
HoxE cluster	Chr5	CM022395.1	Sc28yqQ_31398	Scaffold_17368
HoxA cluster	Chr14	CM022404.1	Sc28yqQ_3523	Scaffold_17368
HoxD cluster	Chr7	CM022397.1	Sc28yqQ_31367	Scaffold_15738
HoxC cluster	Chr14	CM022404.1	Sc28yqQ_3523	

All the information has been updated in the main figures, and provided in Supplementary information S7.

10. The same improvements could be applied to other homeobox families, but I understand that this would imply a lot of work and it is beyond the scope of this manuscript, so I do not think it is necessary to do these analyses with the rest of homeobox genes (other ANTP, PRD, TALE etc). But, as stated by the authors, Hox clusters are iconic for WGDs and it is important that the results related to them are sufficiently clear and detailed.

Response 10:

We have only included the naming of HoxA-E clusters to make consistency with the published Shingate et al study.

11. 3. Minor comments

A. As a general comment, the resolution of the figures is very low, so most of the text included within the figures is impossible to read (scaffolds numbers etc.). I have checked the figures and tried enlarged them by zooming in both in the merged pdf and in the original doc file provided by the authors, but in both of them the resolution was equally low.

Response 11:

We have now adjusted the figures accordingly.

12. B. I understand the paper by Shingate et al NatComm 2020 was probably published after Nong et al submitted the current manuscript, but now that it has been published I think it is important that the authors reference this work as they did with other horseshoe crab genome papers. Similarly, in addition to Nossa et al, it is also important to cite the other publication reporting a *Limulus* genome assembly: Battelle et al GBE 8, 1571–1589 (2016), especially taking into account that the assembly reported by Battelle et al is by far the best publicly available *Limulus* assembly.

Response 12:

The Nossa et al 2014 study was originally cited in the original article, but regretfully got edited out. This has now been remedied. Shingate et al 2020 and Battelle et al 2016 are now both included, and specifically highlighted in the newly added Table 1.

13. C. The methods do not include a section describing how the authors searched for homeobox genes in *Limulus* and which *Limulus* assembly version was used (perhaps the one by Battelle et al GCF_000517525.1 ??). I think it is important to detail this information.

Response 13:

We searched for *Limulus* homeobox genes using the Batelle et al 2016 genome assembly (GCF_000517525). This information is now made clear in the methods section, and shown as follows:

“Homeobox genes in the *Limulus* genome assembly (GCF_000517525) was obtained from previous study (Batelle et al 2016).”

14. D. Table in Fig1. Since Nong et al manuscript compares information from all available xiphosuran genomes and will be a key reference in this regard, I think it would be good to add to this table the stats and accession numbers of the other *Carcinoscorpius* assembly (the one by Shingate et al) and the *Limulus* assembly from Battelle et al that I mentioned in my previous comments.

Response 14:

We agree, and have now added a new Table 1 from the original Figure 1c as suggested, and shown as follows:

Common name	Mangrove horseshoe crab	Mangrove horseshoe crab	Tri-spine horseshoe crab	Tri-spine horseshoe crab	Tri-spine horseshoe crab	Atlantic horseshoe crab	Atlantic horseshoe crab
Species name	Carcinoscorpius rotundicauda	Carcinoscorpius rotundicauda	Tachyplesus tridentatus	Tachyplesus tridentatus	Tachyplesus tridentatus	Limulus polyphemus	Limulus polyphemus
Accession number	WCHO00000000	GCA_011833715.1	WCHN00000000	CNA0000821	QXHF01000000	GCF_000517525	PRJNA187356
Number of scaffolds	30,138	728	39,367	204	671,877	286,793	896,522
Assembly size	1,725,596,044	1,668,682,064	1,718,441,268	2,167,470,406	1,942,936,674	1,828,271,751	1,229,280,963
Gap content	0.85%	0.05%	3.10%	0.17%	1.55%	6.70%	12.65%
Scaffold N50	90,264,435	102,344,000	109,788,719	169,002,194	2,761,313	254,089	2,929
Contig N50	437,918	7,608,042	2,961,265	1,689,442	52,179	11,441	460
Number of genes	34,354	25,985	42,906	34,966	29,134	26,905	/
Complete BUSCOs	93.80%	94.80%	93.70%	95.00%	96.20%	/	/
Reference	This study	Shingate et al 2020	This study	Gong et al 2019	Liao et al 2019	Battelle et al 2016	Nossa et al 2014

15. E. In line 113 the authors say “Furthermore, 89.8% of the sequences assembled for *C. rotundicauda* genome are contained on just 16 pseudomolecules, consistent with a near chromosome-level assembly (chromosome 2n=32, Iswasaki et al 1988, Supplementary information S1, Table 1.1.3)”. Could the authors provide the equivalent information about pseudomolecules and karyotype of the other genome assembly reported in the manuscript, *T. tridentatus*?

Response 15:

Equivalent information is now provided in the Supplementary information S1 Figure S1.1.2, Table 1.1.4.

16. F. In line 167 the authors comment on the different number of Hox clusters found by Gong et al: “In a recent study of the *T. tridentatus* re-sequenced genome, the authors could only find two Hox clusters and could not identify the *Ftz* gene inside these clusters (Gong et al 2019). On contrary, our results suggested that there are three Hox clusters (including *Ftz*), and thus more than one round of WGD occurred in the lineage leading to extant horseshoe crabs.” Could the authors elaborate further on these differences? Are these genes missing in Gong et al’s assembly because of incompleteness? Or are these genes present in the assembly but they were missed by Gong et al because they are not annotated? Have the authors tried to searched for these genes in Gong et al’s assembly using blast? I think it would be important for some readers to understand the source of these discrepancies. The same applies to *Carcinoscorpius*, it would be good if

the authors could clarify why they found more Hox genes that were missed in Shingate et al paper.

Response 16:

Based on the suggestions, we have carried out Blast searches of the Ftz gene in the Gong et al's genome assembly. Interestingly, there are indeed three Ftz genes in the genome assembly just like ours, which seemed to have somehow missed in their analyses in their published paper. The information is shown below:

Species	Scaffold::position_frame /_otherexons	Gene na	Family	Class	Homeodomain sequence	Hox genes in the Gong's genome
T. tridentatus	Scaffold_15738;HRSCAF=16415::18095067_-; Ftz D	Hox6-8	ANTP	PKRTRQYYSRYQTLLEKEFHFNRYLTRRRRIEIAHSLGLTERQKIWFQNRMMKAKK	Hic_chr_9&stop=3661032&start=36610144	
T. tridentatus	Scaffold_44553;HRSCAF=45873::97329232_+; Ftz B	Hox6-8	ANTP	SKRTRQTYTRYQTLLEKEFHFNRYLTRRRRIEIAHSLGLTERQKIWFQNRMMKAKI	Hic_chr_2&start=18639451&stop=18639630	
T. tridentatus	Scaffold_17368;HRSCAF=18114::91043941_-; Ftz A	Hox6-8	ANTP	PKRTRQTYTRYQTLLEKEFHFNRYLTRRRRIEIAHSLGLTERQKIWFQNRMMKQKI	Hic_chr_11&start=27791170&stop=27791349	

We have now annotated the Hox genes as named in the Shingate et al's genome assembly and paper, and the results are updated in the main figures as well as provided in Supplementary information S7 (sheets 2 and 3).

17. G. Supplementary figure with miRNAs alignments. In some cases only the scaffold numbers are indicated and the abbreviation used for each of the species is missing. Please, revise.

Response 17:

The figure has been revised accordingly.

18. H. I found quite interesting that 5 of the novel xiphosuran miRNAs families have several duplicates, these are Novel 2, Novel 3, Novel 4, Novel 5 and especially Novel 10 (which has 4 copies). Could these duplicated miRNAs be considered as duplicated rare genomic changes that support an independent origin of the horseshoe crab WGDs, as I discussed above in my first comment? Are these duplicates found in tandem or in ohnologous regions? It would be interesting that the authors discuss this in the manuscript.

Response 18:

We have now re-analysed the novel miRNAs of both horseshoe crab species and further looked into their conserved loci in the *Limulus* genome as suggested. 7 xiphosuran novel miRNAs are found to be conserved in all three horseshoe crab genomes, while another 13 miRNAs are conserved in both Cr and Tt (Supplementary information S1 Figure S1.2.11). Considering that these xiphosuran miRNAs paralogues are located on different scaffolds in their respective genomes, they are less likely to be generated via tandem duplications. The following sentences have now been added back in to the manuscript on Page 9, lines 20-29 as follows:

“In addition to relatively ancient conserved microRNAs, we also investigated novel microRNAs specific to a certain horseshoe crab species, to understand the impact of WGD on these. We found that 7 xiphosuran novel miRNAs are conserved in all three horseshoe crab

genomes, while another 13 miRNAs are conserved in both *C. rotundicauda* and *T. tridentatus* (Supplementary Figure S1.2.11). Considering that these xiphosuran miRNAs paralogues are located on different scaffolds in their respective genomes, they are unlikely to be generated via tandem duplications. The identified novel microRNAs show higher sequence conservation between orthologues than paralogues (e.g. Supplementary information S1, Supplementary Figure S1.2.11), suggesting these horseshoe crab-specific novel microRNAs originate in the horseshoe crab ancestor following WGD.”

19. I. In line 294 the authors state: “As these genomes have undergone WGD, to confidently reveal their population structure, we only mapped sequencing reads to the mitochondrial genome and constructed the evolutionary trees from mitochondrial data.”

I do not see the relationship between the presence of ancient WGDs and the impossibility of studying populations using nuclear data, if that were really the case, hundreds of population studies performed in vertebrates using nuclear data could not exist. I do not have any problem with the authors’ decision to focus on mitochondrial data, but this statement does not make sense. I suggest to delete it or to use a more convincing argument.

Response 19:

The reviewer is correct, and we have now rephrased the sentence on Page 10 Line 19-20 as follows:

“We firstly mapped sequencing reads to the mitochondrial genome and constructed evolutionary trees from mitochondrial data to identify population structure.”

Reviewers' Comments:

Reviewer #1:

Remarks to the Author:

I am impressed by the revision and applaud the authors to their manuscript.

Reviewer #2:

Remarks to the Author:

The manuscript has greatly improved, I thank the authors for addressing my comments and those of the other referee. I only have two minor comments:

1. Section on homeobox genes in different horseshoe crab species (pages 5-7). Apart from the assemblies they generated, the authors reanalysed previously published xiphosuran assemblies, including one of *Limulus*. If understood correctly, in the revised version of the Methods, the authors indicate that the *Limulus* assembly they used is the one published by Batelle et al in 2016. However, throughout the entire section, the authors use a wrong reference from a different *Limulus* assembly that was not used in the current manuscript (the assembly published by Nossa et al, 2014). Please, use the appropriate reference (Batelle et al, 2016).

2. In lines 15-19 the authors mention that additional data would be required to differentiate between different scenarios regarding the origin of chelicerate WGDs. However, the discussion is unbalanced, since they only mention one of the possibilities: that a WGD was shared between xiphosurans and arachnoplumonates, discussing how this shared WGD would fit into two different phylogenetic frameworks. So, I think the authors should also explicitly mention here the alternative hypothesis, which so far has received a stronger support from existing data: that the WGDs in xiphosurans and arachnoplumonates originated independently.